# **Non-methane organic gas emissions from biomass burning:**

# 2 identification, quantification, and emission factors from PTR-

# **3 ToF during the FIREX 2016 laboratory experiment**

Abigail R. Koss<sup>1,2,3\*</sup>, Kanako Sekimoto<sup>1,2,4</sup>, Jessica B. Gilman<sup>2</sup>, Vanessa Selimovic<sup>5</sup>, Matthew M.
Coggon<sup>1,2</sup>, Kyle J. Zarzana<sup>1,2</sup>, Bin Yuan<sup>1,2,†</sup>, Brian M. Lerner<sup>1,2,‡</sup>, Steven S. Brown<sup>2,3</sup>, Jose L.
Jimenez<sup>1,3</sup>, Jordan Krechmer<sup>1,3,‡</sup>, James M. Roberts<sup>2</sup>, Carsten Warneke<sup>1,2</sup>, Robert J. Yokelson<sup>5</sup>,

Joost de Gouw<sup>1,2,3</sup>

1. Cooperative Institute for Research in Environmental Sciences, University of Colorado Boulder, Boulder,
 CO, USA

- 2. NOAA Earth System Research Laboratory, Chemical Sciences Division, Boulder, CO, USA
- 3. Department of Chemistry, University of Colorado Boulder, Boulder, CO, USA
- 4. Graduate School of Nanobioscience, Yokohama City University, Yokohama, Japan
- 5. Department of Chemistry and Biochemistry, University of Montana, Missoula, MT, USA
- \* Now at Department of Civil & Environmental Engineering, Massachusetts Institute of Technology,
- Cambridge, MA, USA
- † Now at Laboratory of Atmospheric Chemistry, Paul Scherrer Institute, Villigen, Switzerland
- ‡ Now at Aerodyne Research, Inc., Billerica, MA, USA

#### 18 Abstract

Volatile and intermediate-volatility non-methane organic gases (NMOGs) released from biomass 20 burning were measured during laboratory-simulated wildfires by proton-transfer-reaction time-of-flight 21 mass spectrometry (PTR-ToF). We identified NMOG contributors to more than 150 PTR ion masses using gas chromatography (GC) pre-separation with electron ionization, H<sub>3</sub>O<sup>+</sup> chemical ionization, and NO<sup>+</sup> 22 23 chemical ionziation, an extensive literature review, and time-series correlation, providing higher certainty 24 for ion identifications than has been previously available. Our interpretation of the PTR-ToF mass spectrum 25 accounts for nearly 90% of NMOG mass detected by PTR-ToF across all fuel types. The relative 26 contributions of different NMOGs to individual exact ion masses are mostly similar across many fires and 27 fuel types. The PTR-ToF measurements are compared to corresponding measurements from open-path 28 Fourier transform infrared spectroscopy (OP-FTIR), broadband cavity enhanced spectroscopy (ACES), and 29 iodide ion chemical ionization mass spectrometry (I<sup>-</sup> CIMS) where possible. The majority of comparisons 30 have slopes near 1 and values of the linear correlation coefficient,  $R^2$ , of >0.8, including compounds that 31 are not frequently reported by PTR-MS such as ammonia, hydrogen cyanide (HCN), nitrous acid (HONO), 32 and propene. The exceptions include methylglyoxal and compounds that are known to be difficult to 33 measure with one or more of the deployed instruments. The fire-integrated emission ratios to CO and 34 emission factors of NMOGs from 18 fuel types are provided. Finally, we provide an overview of the chemical characteristics of detected species. Non-aromatic oxygenated compounds are the most abundant. 35

Furans and aromatics, while less abundant, comprise a large portion of the OH reactivity. The OH reactivity,
 its major contributors, and the volatility distribution of emissions can change considerably over the course
 of a fire.

01 a me.

#### 39 1. Introduction

40 Biomass burning, including wildfires, agricultural burning, and domestic fuel use, is a large source 41 of non-methane organic gases (NMOGs) to the atmosphere (Crutzen and Andreae, 1990; Akagi et al., 2011). 42 These compounds can be directly harmful to human health (Naeher et al., 2007) and contribute to the 43 formation of secondary pollutants including ozone and secondary organic aerosol (SOA) (Alvarado et al., 44 2009; Yokelson et al., 2009; Jaffe and Wigder, 2012; Alvarado et al., 2015). Because NMOGs from biomass 45 burning are a complex mixture of many species that can change considerably depending on fuel and fire 46 characteristics, many modeling and inventory efforts have had difficulty capturing subsequent chemistry in 47 fire plumes (Alvarado et al., 2009;Grieshop et al., 2009;Wiedinmyer et al., 2011;Heald et al., 2011;Müller 48 et al., 2016; Reddington et al., 2016; Shrivastava et al., 2017). Additionally, a substantial portion of gas-49 phase carbon may be missing from many field measurements (Warneke et al., 2011;Yokelson et al., 50 2013;Hatch et al., 2017) and the gas phase-precursors of SOA are not sufficiently understood (Jathar et al., 51 2014: Alvarado et al., 2015: Hatch et al., 2017). For these reasons, it is important to develop and understand 52 analytical techniques that quantify a large number of biomass burning NMOGs.

Gas chromatography (GC) techniques have been used to identify NMOGs emitted by biomass burning in high chemical detail, and provide exact isomer identifications (Hatch et al., 2015;Gilman et al., 2015;Hatch et al., 2017). However, on-line GC techniques do not provide continuous measurement and are limited to certain classes of NMOGs depending on the column(s) selected and required sample preconditioning steps. This makes them non-ideal for some important compounds or situations where fast, continuous measurements are necessary. Whole-air sampling followed by GC can improve the time resolution, but is affected by artifacts from canister storage (Lerner et al., 2017).

Proton-transfer-reaction mass spectrometry (PTR-MS) is a complementary technique widely used 61 in atmospheric chemistry, both standalone and with a GC interface (de Gouw and Warneke, 2007; Yuan et 62 al., 2017). This chemical ionization technique uses  $H_3O^+$  to detect a wide range of unsaturated and polar 63 NMOGs. It can measure continuously at a very fast rate: up to 10Hz measurement has been reported in the 64 literature (Müller et al., 2010). Recently, PTR-MS instruments using time-of-flight mass analyzers (PTR-65 ToF) with mass resolution greater than 4000 m/ $\Delta$ m have provided fast, simultaneous measurements of exact mass and elemental formula over a wide mass range (m/z typically between 10-500 Th) with detection 66 67 limits in the tens to hundreds of parts-per-trillion (pptv) range (Jordan et al., 2009; Yuan et al., 2016). The

addition of a GC interface can resolve isomers with the same elemental formula thereby providing the exact
 identity of detected NMOGs.

Several recent papers have reported use of high-resolution PTR-ToF to measure biomass burning NMOGs in the laboratory (Stockwell et al., 2015;Bruns et al., 2017) and the environment (Brilli et al., 2014;Müller et al., 2016). Hatch et al. (2017) suggest that PTR-ToF measures a substantial fraction (50-80%) of total NMOG carbon mass. The mass spectra resulting from PTR-ToF detection of biomass burning NMOGs are complex, and many peak assignments are tentative. However, it is clear that PTR-ToF can provide detailed NMOG measurements relevant to studying the effects of fire emissions on human health, and ozone and secondary organic aerosol formation.

A PTR-ToF instrument (Yuan et al., 2016) was deployed during the Fire Influence on Regional and 78 Global Environments Experiment (FIREX) 2016 intensive at the US Forest Service Fire Sciences 79 Laboratory in Missoula, Montana. This experiment burned a series of natural fuels and characterized the 80 gas- and particle-phase emissions with a range of instrumentation (Selimovic et al., 2017). Aging of these 81 emissions was explored with additional chamber experiments (described elsewhere). In this paper we 82 describe the PTR-ToF instrument operation and interpretation of measurements. The focus is on direct 83 emissions. Building on work by Stockwell et al. (2015);Brilli et al. (2014), and others, we provide new, 84 more detailed, and more highly time-resolved chemistry of NMOG emissions from biomass burning than 85 previously available.

The purposes of this work are to improve our understanding of the complex NMOG emissions from 87 biomass burning by interpreting PTR-ToF measurements of biomass burning emissions; provide emission 88 factors and emission ratios to CO for many NMOGs; link PTR-ToF measurements to GC, Fourier-transform 89 infrared spectroscopy (FTIR), and Iodide CIMS (I CIMS) measurements; and report instrument operation 90 and data quality assurance information that will support future analyses. Novel tools to study NMOGs 91 measured by PTR-ToF applied in this work include (1) use of a GC interface to provide an additional level 92 of chemical information; (2) use of NO<sup>+</sup> CIMS (switchable-reagent-ion) chemistry to support compound 93 identification; and (3) use of an improved method to estimate the instrument sensitivity to NMOGs not 94 directly calibrated.

#### 95 2. Methods

#### 96 2.1 Fire Sciences Laboratory experimental setup

Controlled biomass combustion experiments were conducted in a large (12.5m × 12.5m × 22m
high) indoor facility at the US Forest Service Fire Sciences Laboratory. Fuels were burned underneath a
1.6m diameter exhaust stack. Emissions were vented through the stack to 17m height, where a sampling

platform is located. The pressure, temperature, and relative humidity of the air in the combustion chamber 101 were monitored and low light conditions were present during experiments. The Fire Sciences Laboratory 102 facility is described in more detail elsewhere (Christian, 2003, 2004; Burling et al., 2010; Stockwell et al., 103 2014). The FIREX 2016 intensive burned fuels characteristic of the western US, including Ponderosa pine, 104 Lodgepole pine, Douglas fir, Engelmann spruce, subalpine fir, manzanita, chamise, sage, and juniper. 105 Several other types of fuels were also burned, but with fewer replicates, and these included additional pine 106 species (Loblolly and Jeffrey pine), bear grass, rice straw, ceanothus, dung, peat, excelsior, and commercial 107 lumber. Experiments with western fuels included combustion of both specific components of the fuel, such 108 as canopy, litter, and duff, and more realistic burns that included a mix of components (Selimovic et al., 109 2017).

Two types of combustion experiments were conducted. In the first set of experiments, the "stack 111 burns", emissions were entrained into the ventilation stack and measured from the 17m sampling platform. 112 These experiments allowed characterization of changes in emission composition during the course of a fire 113 and typically lasted five to twenty minutes. In the second set of experiments, the "room burns", emissions 114 were not vented and were allowed to mix and fill the combustion chamber. These experiments lasted several 115 hours and provided a more compositionally stable mixture for instruments requiring a longer sampling time. 116 In this work, we discuss 58 stack burns measured directly with PTR-ToF, and these data were used for the 117 comparison between instruments. We also reference measurements from an additional seven stack burns 118 measured directly with NO<sup>+</sup>-CIMS, and discuss results from three stack burns and three room burns that 119 were measured with GC-PTR-ToF, using both  $H_3O^+$  and  $NO^+$  reagent ion chemistry. The particular fires 120 measured with each technique were selected as follows. At least one fire of each fuel type was measured 121 directly with PTR-ToF, and coniferous fuels were measured at least twice. Given these restrictions with the 122 PTR-ToF measurement, the widest possible range of fuel types was measured with NO<sup>+</sup>-CIMS. GC-CIMS 123 stack burns were selected for longer-burning fuels that allowed collection of more than one sample. GC-124 CIMS room burns were selected to explore a range of fuel types. We were not able to measure every fuel 125 type with every instrumental technique. Because there was not a clear temporal separation between fire 126 processes, and because some compounds were lost to the chamber walls (Stockwell et al., 2014), room 127 burns measured directly with PTR-ToF were not used for compound identification and calculation of 128 emission factors.

#### 129 **2.2 Instrumentation**

An overview of the instruments referenced in this work is given in Table 1.

#### 131 2.2.1 PTR-ToF and NO<sup>+</sup>-CIMS

The PTR-ToF is a chemical ionization mass spectrometer typically using  $H_3O^+$  reagent ions. Trace gases with a proton affinity higher than that of water are protonated in a drift tube region, and are detected sensitively with typical detection limits in the tens to hundreds of parts-per-trillion (pptv) range for a 1-sec measurement time. The main advantages of this technique are a response to a wide range of polar and unsaturated NMOGs, a low degree of fragmentation, and fast, on-line measurement capability. PTR-ToF additionally detects several inorganic species, including ammonia (NH<sub>3</sub>), isocyanic acid (HNCO), hydrogen sulfide (H<sub>2</sub>S), and nitrous acid (HONO), which are included in our discussion of NMOGs.

The instrument used in this work is very similar to that described by Yuan et al. (2016), with two 140 relevant differences. The PTR-ToF instrument described by Yuan et al. (2016) includes two RF-only 141 segmented quadrupole ion guides between the drift tube and time-of-flight mass analyzer; while the current 142 version has only one ion guide. The effects of this are that the sensitivities are slightly higher ( $\sim 25\%$  on 143 average), low ion masses (<m/z 40 Th) are transmitted with higher efficiency, and the humidity dependence 144 of NMOG sensitivity is less severe. There is also a higher flow rate (150 sccm) into the drift tube. Second, 145 the instrument inlet (held at 40°C) consists of 1/16" ID PEEK tubing rather than 1/8" PFA, which reduced 146 residence time in the inlet.

The PTR-ToF is equipped with a switchable reagent ion source that allows for  $H_3O^+$  and 147 148 alternatively NO<sup>+</sup> ionization, by flowing either water vapor for H<sub>3</sub>O<sup>+</sup> or ultrapure air for NO<sup>+</sup> through a 149 hollow cathode ion source and adjusting ion source and ion guide voltages. NO<sup>+</sup> chemical ionization of 150 NMOGs creates different product ions than  $H_3O^+$  chemical ionization, and the ionization mechanism 151 depends on functional group (Koss et al., 2016). The PTR-ToF instrument in NO<sup>+</sup> configuration (NO<sup>+</sup>-152 CIMS) can therefore detect several additional classes of NMOGs (e.g. branched alkanes) and can 153 differentiate some sets of isomers, such as aldehydes and ketones, and nitriles and pyrroles. NO<sup>+</sup>-CIMS is 154 described in detail by Koss et al. (2016). The NO<sup>+</sup>-CIMS was used to measure emissions directly, from a 155 small number of coniferous fuels, and as the detector for the GC instrument, for several fuel types.

In  $H_3O^+$  mode the PTR-ToF was operated with an electric field to number density ratio (E/N) of 120x10<sup>-17</sup> V cm<sup>2</sup> (= 120 Townsend or Td). Measurements were made at 2Hz frequency. Ion m/z from 12-157 158 500 Th were measured, and 12-217 Th were quantified with a maximum resolution of 4500 FWHM m/ $\Delta m$ . 159 This is sufficient to resolve many isobaric species, but many peaks still overlap in the mass spectrum. 160 Overlap of an ion peak by an intense neighbor can strongly affect the accuracy of that ion measurement, 161 and such affected ions were excluded from further analysis. ToF data were analyzed using the Tofware 162 software package (Aerodyne Research Inc./ Tofwerk AG). For approximately one-half of the stack 163 experiments, NMOG ion concentrations were temporarily high enough to deplete the reagent ion by 10-164 50%. Under these conditions, sensitivity to NMOGs is lower and nonlinear (Veres et al., 2010b). We 165 corrected NMOG ion signals for this effect, although effects from secondary proton-transfer reactions could 166 still be a significant source of inaccuracy (Section S1). Raw ion count rates (counts-per-second, cps) were 167 corrected for duty-cycle discrimination in the ToF extraction region, and normalized to the intensity of the 168 reagent ion ( $H_3O^+$  10<sup>6</sup> cps or NO<sup>+</sup> 10<sup>6</sup> cps). Correction for humidity effects and conversion of ion signal to 169 mixing ratio are discussed in Section 2.3. Before each fire, we first measured instrument background by 170 passing air from the combustion chamber through a heated platinum catalyst, then measured chamber 171 background. Concentrations of NMOGs during the fire were generally several orders of magnitude higher 172 than either background.

The NO<sup>+</sup>-CIMS/PTR-ToF transfer inlet was <sup>1</sup>/<sub>2</sub>" OD (3/8" ID) PFA, heated to 40°C, with a flow rate of 100 SLPM. It was 16m long (residence time 0.7 seconds) and located on the sampling platform for stack burns, and 7m long (residence time 0.3 seconds) and located 3m above the combustion chamber floor for room burns. The instrument subsampled 500 sccm through a 40°C 10 cm 1/16" ID PEEK capillary orthogonally via PFA branch reducing tee mounted to the main inlet. Most particles were separated from the CIMS subsample capillary through virtual impaction, although a small, unquantified amount of particulate matter did enter the smaller instrument inlet.

NMOGs could be lost to transfer inlet, instrument tubing, or drift tube surfaces. Based on good 181 agreement with instrumentation on the sampling platform (Section 3.3.1) inlet losses of highly volatile 182 compounds were negligible, but we were not able to quantify possible losses of less volatile compounds. 183 Measurement of compounds with saturation vapor pressure ( $C_0$ ) less than 10<sup>5</sup> µg m<sup>-3</sup> may be affected 184 (Pagonis et al., 2017). Slight delay in the instrument response to some compounds with  $C_0$  close to 10<sup>4</sup> µg 185 m<sup>-3</sup> was observed.

#### 186 2.2.2 GC-MS and GC-PTR-ToF

The gas chromatograph (GC) instrument cryogenically pre-concentrates 4-minute samples of NMOGs before separation on one of two capillary columns (Lerner et al., 2017). The sample stream is separated into two channels that are optimized to reduce water and carbon dioxide before cryogenic trapping of NMOG. The first channel (trapping at -165 °C) is connected to an Al<sub>2</sub>O<sub>3</sub>/KCl Porous Layer Open Tubular (PLOT) column optimized for C<sub>2</sub>-C<sub>6</sub> hydrocarbons. The second channel (trapping at -135 °C) uses a medium polarity polysiloxane (Restek MXT-624) column optimized for C<sub>6</sub>-C<sub>10</sub> hydrocarbons and many polar compounds. The two channels are analyzed sequentially.

The eluent from the GC columns was directed to either an electron ionization (EI) quadrupole mass spectrometer (Agilent model 5975C) or to the PTR-ToF. The quadrupole mass spectrometer has unit mass resolution and was operated in full ion scan mode, from m/z 19 to 150 Th. When the PTR-ToF, in either  $H_3O^+$  and NO<sup>+</sup> configuration, was used as the detector, the 2 sccm eluent from the columns was introduced directly into the drift tube. To maintain pressure (2.4 mbar) in the drift tube, an additional 50 sccm of catalyst-generated clean air was added to the drift tube. This is lower than the 150 sccm of flow used during non-GC-PTR-ToF operation but does not affect compound identification.

- The GC inlet for stack burns was  $\frac{1}{2}$ " OD PFA, 16m long, located on the sampling platform, with a continuous flow rate of 20 lpm. A subsample was directed to the instrument with a  $\frac{1}{4}$ " OD PFA, 2m long line with flow rates from 2-7 lpm. For room burns the inlet was  $\frac{1}{4}$ " OD PFA, 7m long, and located 3m above the combustion chamber floor. A flow rate of 2-7 SLPM was used. For both stack and room burns the inlet was heated to 40°C and the stream was dynamically diluted with humidified UHP N<sub>2</sub> (1 to 3 parts smoke to 5 parts N<sub>2</sub>). Particles were reduced by virtual impaction.
- Two stack experiments (both Douglas fir) were measured with both GC-EI-MS and GC-PTR-ToF; 208 one stack experiment (Englemann spruce duff) and three room experiments (Douglas fir, Subalpine fir, and 209 sage) were each measured with GC-EI-MS, GC-PTR-ToF, and GC-NO<sup>+</sup>-CIMS. Two additional samples 210 (of a room burn of sage), one with  $H_3O^+$  and one with NO<sup>+</sup> chemistry, were analyzed using an accelerated 211 GC temperature ramp program to better observe late-eluting compounds. Each four-minute sample was 212 analyzed with just one type of detector, and the detector was switched for the next four-minute sample. For 213 room experiments and duff stack burns, NMOG composition was largely consistent between successive 214 four-minute GC samples. Other stack burns varied more quickly. The room experiment GC-CIMS analyses 215 detected NMOGs more sensitively because we were better able to adjust the GC sample stream dilution. 216 Finally, we measured a 56-component NMOG calibration standard with the GC-PTR-ToF and GC-NO+-217 CIMS (three replicates) to help establish GC retention times.

#### 218 **2.2.3 Other instrumentation**

A number of trace gases measured by the PTR-ToF were also measured by other instruments (Table 1), and in Section 3.3 we compare these measurements. The OP-FTIR instrument was located on the sampling platform with the optical path spanning the stack, and therefore did not have an inlet (Stockwell et al. 2014). The OP-FTIR employed a time resolution of 1.37 seconds and the PTR-ToF data were interpolated to the OP-FTIR sampling times for the intercomparison.

Glyoxal, methylglyoxal and HONO were measured with the NOAA Airborne Cavity Enhanced Spectrometer (ACES) instrument, which uses broadband cavity enhanced spectroscopy. Wavelength resolved gas-phase extinction was measured in two spectral regions, one in the UV (361 nm to 390 nm) and one in the blue (438-468 nm), and then fit using literature cross sections to retrieve the concentrations of NO<sub>2</sub>, HONO, methylglyoxal, and glyoxal (Min et al., 2016). Data from this instrument were reported at 1second intervals. The ACES instrument inlet was located on the sampling platform, with an inlet of approximately 1m length sampling from the center of the stack flow directly above the OP-FTIR opticalpath.

The I- CIMS chemically ionizes organic and inorganic gases through iodide adduct formation, and analyzes the resulting ions with a high-resolution time-of-flight mass spectrometer (Lee et al., 2014). The I CIMS instrument shared an inlet with the PTR-ToF. Air was subsampled from this inlet and dynamically diluted with UHP N<sub>2</sub> to prevent reagent ion depletion. The dilution factor was determined by comparing the CO<sub>2</sub> concentration before and after dilution measured by a LICOR LI6252 co-located with the I<sup>-</sup> CIMS. I<sup>-</sup> CIMS calibration factors were determined by direct calibration for the species discussed here.

#### 238 **2.3 Calibrations and method for estimating calibration factors**

The calibration factor (units of ncps/ppby) is the normalized counts per second (ncps) per ppby of 240 the NMOG(s) whose PTR product is that ion. The ncps are derived from the raw ion count rate (counts per 241 second, cps), corrected for the mass-dependent duty-cycle of the ToF extraction, and normalized to the 242 detected ion count rate of the primary ion (H<sub>3</sub>O<sup>+</sup> cps  $\times 10^{-6}$ ). We detect 8.5-11.5 $\times 10^{6}$  H<sub>3</sub>O<sup>+</sup> ions per second. 243 We detect about 1000 cps/ppbv of acetone and 650 cps/ppbv of benzene. We provide the sensitivity here 244 as raw ion count rate to enable comparison to other PTR-MS instruments, which may have different 245 detected intensity of  $H_3O^+$ . This is about an order of magnitude higher than similar generation commercially 246 available PTR-ToF (Jordan et al., 2009), but an order of magnitude lower than new PTR-ToF instruments 247 that use a different drift tube design (Breitenlechner et al., 2017). Calibration factors in this work were 248 obtained by (1) direct calibration, (2) calculation using kinetic rate constants (Sekimoto et al., 2017a), or 249 (3) comparison with OP-FTIR, which will be discussed in section 3.3. Calibration factors for all ion masses 250 are provided in Tables S1 and S10.

The calibration factors of 37 species were determined experimentally by introducing a known concentration of a NMOG from a standard cylinder, a permeation source (Veres et al., 2010a), a diffusion cell for isocyanic acid and methyl isocyanate (Roberts et al., 2010), or a liquid calibration unit (Ionicon Analytik). The calibration factors of these species have an error of 15% (details in Table S1).

It is unrealistic to experimentally determine calibration factors for all NMOG species detected in biomass burning. Many compounds are highly reactive and cannot be purchased from a commercial supplier. Several methods to estimate calibration factors have been previously used by PTR-MS operators. For example, both Warneke et al. (2011) and Stockwell et al. (2015) estimated calibration factors for uncalibrated species based on ion mass to charge ratio and chemical formula in the latter case.

Sekimoto et al. (2017a) recently developed an improved method of estimating calibration factors. 261 The instrument calibration factor is linearly proportional to the kinetic capture rate constant of the  $H_3O^+$ 262 proton transfer reaction, with additional corrections for mass-dependent transmission and NMOG ion

8

fragmentation, both of which can be constrained experimentally. The proportionality is determined by direct 264 calibration of a small subset of NMOGs. For this work, we used a calibration gas standard containing 265 acetonitrile, acetaldehyde, acetone, isoprene, 2-butanone (methyl ethyl ketone, MEK), benzene, toluene, o-266 xylene, and 1,2,4-trimethylbenzene dynamically diluted to 1-10 ppb. The kinetic capture rate constant can 267 be calculated using the polarizability and permanent dipole moment of the NMOG or alternatively for 268 unidentified ions using the NMOG molecular mass and elemental composition. Figure 1 compares the 269 measured (from this work) and calculated (using the method from Sekimoto et al., 2017a) calibration factors 270 for several compounds. Most calculated calibration factors (72%) fall within  $\pm 10/-50\%$  of the measured 271 sensitivity. The calculated calibration factor provides the upper limit to the sensitivity, and some of the 272 measured calibration factors are lower than predicted. These typically include species with proton affinity 273 close to water (e.g. formaldehyde) and species that fragment to small masses (e.g. ethanol). A detailed 274 discussion of why measured calibration factors can deviate from calculated ones is given in Sekimoto et al. 275 (2017a).

If an identified ion mass has only one NMOG contributing, as is the case for 65% (102) of the ion masses with signal in the fire, we used the calibration factor from direct calibration or the Sekimoto et al. (2017a) method. If an identified ion mass has more than one NMOG contributing, we used a weighted average of the calibration factors of all NMOG contributors to this ion mass (Eq. 1). The determination of relative NMOG contributions to the total ion signal of each individual mass was based on GC-PTR-ToF measurements, comparisons to other instruments, time-series analysis, and reported values from literature and will be described in Section 3.

$$cal factor_{average} = \left(\sum_{i} \frac{contribution_{i}}{cal factor_{i}}\right)^{-1}$$
 Eq. 1

The uncertainty for calibration factors for identified NMOGs ranges from 15% to 50% depending 285 on the calibration method used (Table S1). For ion masses for which we were not able to propose a NMOG, 286 a calibration factor was estimated based on the elemental composition of the ion mass (Sekimoto et al, 287 2016a). The uncertainty for calibration factors for unidentified species is within 10% higher to 50% lower. 288 Ambient humidity can change the measured sensitivity of a NMOG species (Yuan et al., 2016). 289 For species whose calibration factor was measured, a humidity correction factor was also experimentally 290 determined. We currently have no method to predict the humidity dependence of the sensitivity for other 291 species, so for all other species no humidity correction was applied. To minimize the error from this 292 omission, we calibrated compounds that were abundant in emissions and that likely have strong humidity 293 dependence. These include compounds with proton affinities close to water (e.g. HNCO) and compounds 294 whose ionization mechanism includes loss of water (e.g. 1-propanol). Excluding these compounds, the 295 average measured humidity correction factor was less than 15% for the humidity conditions experienced

- during FIREX (5-18 g/kg). Measured sensitivities of different NMOGs both increased and decreased with
- humidity and an unknown humidity correction will likely only cause a small bias for total NMOG signals.
- There were no systematic differences in humidity between fires of different fuels.

#### 299 **3. Results & discussion**

#### 300 **3.1 Identification of PTR-ToF ion masses**

During the Fire Lab experiments we measured 574 ions that were enhanced in emissions from one 302 or more fuel types. Of these, we identified 156 ion masses with a high degree of certainty and for which a 303 calibration factor can be determined. An additional 12 ion masses were identified as fragments of one or 304 more NMOGs whose main product ion was already included in the list of 156 ions. Finally, 4 ions were 305 identified as being a common product of a large number of structurally dissimilar NMOGs. These 172 ions, 306 their identification, and support for that identification are listed in the supplemental information. The 307 supplemental provides detailed information on the isomer contributions to each mass (Table S1), 308 sensitivities and calibration uncertainty(Tables S1, S10), literature references (Table S6), GC measurements 309 (Table S7), and observations from time series correlations (Table S9). The supplemental additionally 310 includes quantitative information on OH rate constants (Table S5), instrument intercomparisons (Table S8), 311 and NO<sup>+</sup>-CIMS ion mass identifications (Table S4). These 172 masses represent about 95% of the total signal (ncps) from m/z 12-217 Th measured by PTR-ToF. Below, we describe the methods used to ascribe 312 313 NMOG identifications to PTR-ToF ion masses.

#### 314 **3.1.1 Literature survey**

Identifications of many NMOGs emitted from biomass burning have been previously reported, 316 using GC, PTR-MS, and optical methods. We compiled a list of observed NMOGs and identifications to 317 use as a starting point. The papers we referenced included Karl et al. (2007), Warneke et al. (2011), Brilli et al. (2014), Stockwell et al. (2015), Müller et al. (2016), and Bruns et al. (2017), which focus on PTR-MS 318 319 measurements, and Gilman et al. (2015), Hatch et al. (2015), and Hatch et al. (2017), which focus on GC 320 measurements. Gilman et al. (2015) used 1D-GC and focused on the most volatile species, and Hatch et al. 321 (2015) and Hatch et al. (2017) used 2D-GC and included many additional less volatile species. NMOG 322 emission factors of identified compounds and the estimated mass of unidentified species have been 323 reviewed by fire/ecosystem type globally (e.g. Akagi et al. (2011); Yokelson et al. (2013)), but significant, 324 recent measurements have not yet been included in the on-line updates: e.g. (Hatch et al., 2017). Finally, 325 for some compounds, we referenced studies of pyrolysis products of lignin, cellulose, and hemicellulose, 326 which used GC-MS, X-ray spectroscopy, FTIR, theoretical calculations, and other analytical methods to

identify major products and common reaction pathways (Patwardhan et al., 2009;Lu et al., 2011;Zhang et
al., 2012; Heigenmoser et al., 2013;Collard and Blin, 2014; Liu et al., 2017a).

We assessed each identification as strongly or weakly supported. Strong identifications include 330 those reported by many separate studies, NMOGs identified using GC methods (especially 2D-GC-ToF-331 MS), and those supported by evidence from pyrolysis or other literature. Weak identifications include those 332 with disagreement between different studies, tentative identifications based on only mass-to-charge ratio or 333 elemental formula, and identifications that are inconsistent with reported formula or that are chemically 334 implausible (e.g. highly strained structure). Identifications from literature and citations are listed in Table 335 S6. Overall we found literature evidence for 68% of our ion identifications. Our interpretation differs from 336 previously published PTR-MS interpretations for 34 ion masses as noted in Table S6. Forty-eight ion 337 masses have not been previously reported in PTR-MS measurements of biomass burning.

The compounds with new and revised identifications were compared to review values of emission factors in Akagi et al. (2011) and Yokelson et al. (2013). A limited number of species from PTR are included in these reviews, largely because of uncertainty in identification. PTR species that have been detected but not included in review tables of EF include many more highly functionalized and larger molecules, and most of our updated identifications are these species. Yokelson et al. (2013) do include a number species from PTR (ion trap) that were not identified, and the identities of many of these have now been determined in this work.

Compounds that are included in review tables, and for which we have updated the assignment are mostly unsaturated hydrocarbons and heteroatom containing species, where the identifications have been updated to include other contributing VOCs. For such species, whose EF was determined solely from PTR, the actual emission factor should be lower than the reported value.

#### 349 **3.1.2 GC-PTR-ToF measurement**

Using gas chromatographic separation before measurement with PTR-MS is a powerful tool that 351 has been widely used in many environments (Warneke et al., 2003;Karl et al., 2007;Warneke et al., 352 2011; Yuan et al., 2014). The combination of measured chromatographic retention time and product ions 353 with GC-PTR-ToF, GC-NO+-CIMS, and GC-EI-MS allows the unambiguous identification of the various 354 isomers contributing to the PTR-ToF signal of many ions. Some additional ion masses had high signal in 355 direct measurement of fire emissions, but did not appear in any chromatographs. This also provides insight 356 into the NMOG chemical structure, as certain functional groups, like acids, cannot travel through the GC 357 system. An example of GC-PTR-ToF measurement is shown in Fig. 2. Panel A in this figure shows the 358 dense chromatographic elution of hundreds of peaks over the 800-second elution period. These 359 chromatographic peaks are detected on several hundred PTR ions. Panel B shows the measured intensity

of m/z 68.050  $C_4H_5NH^+$  during a 280-second segment of the elution, which includes product ions from pyrrole and several butene nitriles. These isomers cannot be distinguished by online PTR-ToF, and each contributes a different amount to the total signal of m/z 68.050  $C_4H_5NH^+$ . Panel C shows the same 280second retention-time period, from a sample taken immediately after the one shown in Panel B, but measured with NO<sup>+</sup>-CIMS. These isomers can be identified by comparing GC-PTR-ToF and GC-NO<sup>+</sup>-CIMS chromatography, as NO<sup>+</sup> reacts with pyrrole but not nitriles. The GC retention time, when it is known for a particular compound, provides additional support for the identification.

The relative intensities of the eluted peaks were used to quantify the relative contribution of each 368 NMOG to each ion mass. The size of a chromatographic peak is determined not only by the mixing ratio 369 of that NMOG in ambient air and the mass spectrometer response, but also by the trapping and elution 370 efficiencies of the GC pre-separation unit. As isomers have the same molecular weight and elemental 371 composition, their volatilities and trapping efficiencies are generally similar. For example, pyrrole and 3-372 butene nitrile have similar vapor pressures of 1.1 and 2.5 kPA at 25°C, compared to ethane (4000 kPA) and 373 1,4-diethylbenzene (0.13 kPA) which are the most and least volatile species measured by the GC, respectively (values from CRC Handbook, 97<sup>th</sup> ed). Here we assumed that all compounds that create the 374 375 same PTR ion mass have similar GC trapping efficiencies. This assumption is supported by GC-PTR-ToF 376 measurements of C4-alkenes, C5-alkenes, C8-aromatics, and C9-aromatics in the 56-component NMOG 377 GC calibration standard. These isomer groups have equal concentrations in the calibration gas and their 378 resulting GC-PTR-ToF chromatographic peaks had similar areas.

The same GC methods were used to identify some signals from the NO<sup>+</sup>-CIMS. Observed and identified NO<sup>+</sup>-CIMS ion masses are included in Table S4. Hundreds of carbon-containing ion masses are also present in a typical NO<sup>+</sup>-CIMS mass spectrum. Using GC-NO<sup>+</sup>-CIMS, we identified the NMOG contributors to an average (across all fires measured with NO<sup>+</sup>-CIMS) of 32% of the total signal of these ions. More identifications could likely be made by analysis with other techniques (intercomparisons, timeseries correlations, literature review, etc.) but were not attempted here. The NO<sup>+</sup>-CIMS ion mass identifications are included here as a reference for future work, but are not discussed further.

#### 386 3.1.3 Time-series correlation

Some species measured by the PTR-ToF have several possible isomers, have not been previously reported in the literature, and are not transmittable through the GC. The identifications of these compounds are less certain. For these, we selected several reasonable isomeric structures based on the types of compounds typically seen in biomass burning emissions: substituted furans and aromatics, nitriles, pyridines, terpenes, and carbonyls. Then, we compared the temporal profile of these ion signals during several fires to compounds with more certain identification. Compounds with similar structure and functionality likely have similar behavior. Dissimilar compounds can also sometimes have similar temporal profiles (Yokelson et al., 1996), but it is still likely that time series correlation points to the correct assignment or a species with similar chemical functionality as the true assignment.

An example of how time-series correlation is used to identify a species is shown in Fig. 3. m/z397 115.039 Th  $C_5H_4O_3H^+$  is the unidentified species, for which there is no strong literature or GC evidence. 398 This isomers. including formula has several plausible furan alcohols (e.g. 399 dihydro(hydroxymethyl)furanone) and methyl-dihydrofurandione. Several other furan alcohols have been 400 including 2-furanmethanol unambiguously identified, (from GC-PTR-ToF) and 2.5-401 (hydroxymethyl)furfural (reported in pyrolysis literature, Lu et al. (2011)). Dihydrofurandione has also 402 been identified (limited isomeric possibilities). Comparing the time series of these species during a stack 403 experiment fire shows that m/z 115.039  $C_5H_4O_3H^+$  correlates better with furan alcohols than with 404 dihydrofurandione. Thus m/z 115.039 is more likely to be a furan alcohol. Based on structural similarity 405 and reported pyrolysis pathways that frequently produce 2.5- substituted furans (Collard and Blin, 2014), 406 dihydro-5-(hydroxymethyl)-2[3H]-furanone is a likely compound.

#### 407 **3.2 NMOG** ion speciation for different fuel types and fire conditions

The contribution of isomers to any particular PTR ion exact mass was consistent between the four 409 fuels (Douglas fir, Engelmann spruce duff, Subalpine fir, and sage) sampled with GC-PTR-ToF (Table S7). 410 Comparing all GC-PTR-ToF samples, the isomeric speciation on a particular exact mass typically varied 411 by only 11% (the standard deviation of the contribution of each isomer to total signal on that mass) and 412 therefore the same study-average NMOG contributions to each ion exact mass were used for all fuel types, 413 whether or not supporting GC information was available. This is similar to the variation of isomer speciation 414 reported by Hatch et al. (2015) (5% on average), who investigated six diverse fuel types. Compounds that 415 had larger variability between GC-PTR-ToF samples (and between fuel types) include m/z  $67.054 \text{ C}_{5}\text{H}_{6}\text{H}^{+}$ 416 (cyclopentadiene), which has substantial and variable interference from an isoprene fragment; and m/z417 153.127  $C_{10}H_{16}OH^+$ , which consists mainly of camphor in sagebrush fires, and of other oxygenated 418 monoterpenes in fires of other fuels. Additionally, in burns of ceanothus, which was not sampled with GC-419 PTR-ToF, m/z 133.065 C<sub>9</sub>H<sub>8</sub>OH<sup>+</sup> was enhanced, did not correlate as well with benzofuran (m/z 119.049 420  $C_8H_6OH^+$ ) and may include a contribution from another isomer such as cinnamaldehyde.

The instantaneous speciation of isomers may also change over the course of a fire, especially as the fire shifts between various higher and lower temperature chemical processes. We used time-series correlation to identify several masses that may have variable NMOG contributors. This analysis was done on Fire 02, which burned representative Ponderosa Pine forest-type fuels. This fire was selected because Ponderosa Pine was the most comprehensively measured fuel type during the FIREX 2016 experiment, this 426 particular fire had distinctly different NMOG speciation at the beginning (higher temperature) and end427 (lower temperature) of the fire, and reagent ion depletion did not affect the results.

We identified three ions with high signal whose NMOG contributors may be substantially different between high- and low-temperature processes in a fire:  $m/z \ 109.065 \ C_7 H_8 OH^+$ , which likely includes more 2-methylphenol from high-temperature processes and more anisol from lower temperature processes; m/z112.039  $C_5 H_5 NO_2 H^+$ , which likely includes a greater contribution from methyl maleimide in high temperature processes and more dihydroxy pyridine from low temperature processes; and  $m/z \ 123.080$  $C_8 H_{10} OH^+$ , which likely includes more C2-phenols from high temperature processes and more methylanisol from low temperature processes (similar to  $m/z \ 109$ ). Time series comparisons are shown in Fig. S2.

These three pairs of identifications in Fig. S2, and their relative contributions to total ion signal, are not well constrained. An additional instrument technique, such as a fast GC capable of separating substituted furans and aromatics, or a better understanding of I<sup>-</sup> CIMS chemical specificity and more accurate calibration on both instruments, would be helpful. To convert instrument signal (ncps) of these ions to mixing ratio, we applied the average calibration factor of the two isomers.

#### 440 **3.3 Intercomparison with other instruments**

Several species detected by the PTR-ToF were also measured by other instruments. The 442 intercomparison is summarized in Fig. 4. All slopes shown in the figure and discussed in the text are the 443 orthogonal distance regression (ODR) slope of  $H_3O^+$ -CIMS to the other instrument; R<sup>2</sup> values are from 444 vertical distance regression of PTR-ToF against the other instrument. The scatter plots are shown in Figures 445 S3-S5.

#### 446 **3.3.1 Comparison with OP-FTIR**

Fifteen species were compared between the PTR-ToF and FTIR (Figure S4). Methanol, 448 formaldehyde, formic acid, propene, acetic acid, ethene, acetylene, furan, phenol, and furfural were 449 calibrated directly on the PTR-ToF and have uncertainty of 15%. For HONO, HCN, and ammonia, we were 450 not able to determine a calibration factor directly and so we set the calibration factors equal to the slope of 451 the comparison between the FTIR and PTR-ToF during Fire 72 (Ponderosa pine with realistic fuel mixture, 452 selected for early data availability, long burning time of 30 minutes, and mix of flaming and smoldering 453 conditions). Sensitivity to HCN has strong humidity dependence (Knighton et al., 2009; Moussa et al., 454 2016), and this was experimentally determined and corrected. Glycolaldehyde was calibrated using the 455 method from Sekimoto et al. (2017a) with an uncertainty of 50%; the PTR-ToF measurement of m/z 61.028 456  $C_{2}H_{4}O_{2}H^{+}$  (sum of glycolaldehyde and acetic acid) has an uncertainty of 27%. FTIR hydroxyacetone was 457 compared to PTR-ToF m/z 75.044  $C_3H_6O_2H^+$ , which was calibrated using the Sekimoto et al. (2017a) method and is the sum of methyl acetate (estimated 37% of mixing ratio), ethyl formate (14%), and
hydroxyacetone (48%), with uncertainty 50%. 1,3-butadiene was calibrated with the Sekimoto et al. (2017a)
method and has an uncertainty of 50%. The method from Sekimoto et al. (2017a) provides the lower bound
of concentration.

Methanol has agreed within stated uncertainties between PTR-MS and FTIR in several previous 463 studies (Christian, 2004;Karl et al., 2007;Warneke et al., 2011;Stockwell et al., 2015), and this work shows 464 an average slope of 0.99 and R<sup>2</sup> of 0.95. The comparison of formaldehyde between PTR-ToF and FTIR has 465 an average slope = 1.1 and average  $R^2 = 0.94$ , which is consistent with the comparison shown in Warneke 466 et al. (2011). Other compounds that compare within the stated uncertainty in slope and have correlation coefficient >0.8 are ammonia, the sum of acetic acid and glycolaldehyde (compared to PTR-ToF m/z 61.028 467  $C_2H_4O_2H^+$ ), formic acid, HONO, acetylene, propene, and HCN. HONO was sufficiently concentrated (900 468 469 ppby max) in the fire, and the precision and accuracy of the FTIR HONO measurement were adequate to 470 estimate a PTR-ToF 3σ LoD for HONO of about 9.5 ppbv. This is likely not sufficient to measure HONO 471 in ambient air except in the most highly concentrated, fresh biomass burning plumes.

The high degree of correlation between PTR-ToF and FTIR for acetylene and ethene is notable, 473 because these two compounds cannot be ionized by proton transfer from H<sub>3</sub>O<sup>+</sup> as their proton affinities are 474 too low. The detected NMOG product ions (acetylene, at m/z 26.015  $C_2H_2^+$ ) and ethene (m/z 28.031  $C_2H_4^+$ ) 475 are most likely the product of charge transfer from contaminant  $O_2^+$  from the ion source, which was high at 476 12% of  $H_3O^+$  during this experiment. The acetylene comparison has a higher degree of scatter ( $R^2 = 0.83$ ), 477 which is likely an effect of interferences from fragments of other species as identified by GC-PTR-ToF. 478 Ethene has better correlation ( $R^2 = 0.94$ ); from the GC-PTR-ToF we observed that m/z 28.031 C<sub>2</sub>H<sub>4</sub><sup>+</sup> is 479 specific for ethene. The disagreement in slope may be due to variability in  $O_2^+$ .

Other compounds including 1,3-butadiene, furan, hydroxyacetone, phenol, and furfural, agreed within a factor of two (slopes of 1.6, 1.5, 0.6, 0.7, and 0.6, respectively) and average R<sup>2</sup> values <0.8. These 481 482 species were often near the 0.73 Hz detection limit of the OP-FTIR and the discrepancy in slopes and low 483 correlation coefficients are sometimes an effect of including this data in the intercomparison. Another 484 reason for the disagreement may be that these species have either more interference or weaker spectral 485 features than other compounds reported from FTIR. Furan may have an interference in PTR-ToF 486 measurements of some fuels (Table S7). Emission ratios and emission factors (EF) are based on fire-487 integrated excess values that benefit from significant signal averaging. Many of the above species have EF 488 that agree between PTR-ToF and FTIR within 10% (Selimovic et al., 2017; Table S8). Additionally, it has 489 been shown that the FTIR fire-integrated emission factors derived for hydroxyacetone is in excellent 490 agreement to that reported for real wildfires by Liu et al., (2017b) (Selimovic et al., 2017).

#### 491 **3.3.2 Comparison with ACES**

Three species were compared between the PTR-ToF and ACES: HONO, glyoxal, and 493 methylglyoxal (Figure S3). HONO agrees with an average slope of 1.13 and  $R^2$ =0.95. Since the PTR-ToF 494 sensitivity factor for HONO was determined by comparison to FTIR, this slope indicates the agreement 495 between FTIR and ACES. Methylglyoxal has a slope of 0.42 and R<sup>2</sup>=0.85. The poorer agreement for 496 methylglyoxal is probably due to interferences on both instruments. The PTR-ToF measures both 497 methylglyoxal and acrylic acid at m/z 73.028 C<sub>3</sub>H<sub>4</sub>O<sub>2</sub>H<sup>+</sup>; both were calibrated using the Sekimoto et al. 498 (2017a) method. The calculation has uncertainty of 50% and gives the lower bound of concentration. The ACES instrument measures a series of substituted  $\alpha$ -dicarbonyls, including 2,3-butadione, from a relatively 499 500 diffuse absorption band that is common to these species. Development of a specific measurement for methyl 501 glyoxal is a target of future research, because this compound is an important SOA precursor whose emission 502 from biomass burning has not been well constrained (Hays et al., 2002;Fu et al., 2008). The methylglyoxal 503 measurement may be improved with changes to the ACES resolution and spectral correction routines.

The comparison of glyoxal is similarly poor (slope =2.56 and  $R^2 = 0.64$ ). This is probably because of incomplete resolution of m/z 59.013 C<sub>2</sub>H<sub>2</sub>O<sub>2</sub>H<sup>+</sup> from m/z 59.049 C<sub>3</sub>H<sub>6</sub>OH<sup>+</sup> (acetone), which is a very large neighboring peak in the PTR-ToF mass spectrum. Poorly resolved peaks such as glyoxal are normally not reported (Section 2.2.1). PTR-MS has been shown to have low sensitivity to glyoxal (LoD=250-700 pptv), with strong humidity dependence, and can be easily lost on inlet surfaces (Stönner et al., 2017). Additionally, the PTR-ToF glyoxal sensitivity was calculated and has an uncertainty of 50%. The glyoxal measurement may be significantly improved with better PTR-ToF sensitivity and mass resolution.

#### 511 **3.3.3 Comparison with I**<sup>-</sup> CIMS

Some data were compared to I CIMS for one fire (Fire 72, Ponderosa pine with realistic blend of 513 fuel); a more detailed comparison will require significant additional analysis of the I<sup>-</sup> CIMS data set. 514 Although many ion masses overlap between the PTR-ToF and I<sup>-</sup> CIMS, we selected seven that have straightforward interpretation on both instruments: HCN, formic acid, phenol, vanillin, acetic acid and 515 516 glycolaldehyde, acrylic acid and methylglyoxal, and cresol and anisol (Figure S5). These compounds were 517 all directly calibrated on the I- CIMS, with an uncertainty of  $\pm 15\%$ . Formic acid, phenol, vanillin, acetic 518 acid, cresol, and anisol were calibrated directly on the PTR-ToF, and the HCN sensitivity was taken from 519 the comparison to FTIR. Glycolaldehyde, acrylic acid, and methylglyoxal were calibrated using the 520 Sekimoto et al. (2017a) method with an uncertainty of 50%. The comparison for HCN, formic acid, and phenol is excellent (slopes = 0.97, 0.94, and 1.08; R<sup>2</sup>=0.99, 0.99, 0.98, respectively). The vanillin 521 522 measurements also agree quantitatively (slope = 0.92), but the I<sup>-</sup> CIMS measurement is noisier ( $R^2 = 0.71$ ). 523 For the other three species, the I<sup>-</sup> CIMS measures only one isomer, while the PTR-ToF measures a sum of 524 several isomers. For all three, the comparison is within the stated uncertainties of both instruments, but the 525 PTR-ToF measurement is lower than the I CIMS measurement. The PTR-ToF measurement of acrylic acid 526 plus methylglyoxal is 31% lower than the I CIMS measurement of acrylic acid; the PTR-ToF measurement 527 of acetic acid plus glycolaldehyde is 17% lower than the I<sup>-</sup> CIMS measurement of acetic acid; and the PTR-528 ToF measurement of cresol plus anisol is 1% lower than the I CIMS measurement of cresol. The low PTR-529 ToF measurement for the acrylic acid and cresol comparison is possibly due to uncertainty in the calculated 530 calibration factors, which give the upper limit to sensitivity (and the lower limit to derived concentration). 531 The acetic acid comparison is within the stated uncertainty (27% for PTR-ToF m/z  $61.028 C_2H_4O_2H^+$  and 532 15% for I<sup>-</sup> CIMS acetic acid).

#### 533 **3.4 Emission factors, emission ratios, and emission chemistry**

We quantified the emission ratios relative to CO, and the emission factors in g/kg fuel burned, of 535 both the identified and unidentified species. The emission ratio (ER) is calculated by Eq. 2:

$$ER = \frac{\int_{t=0}^{t=end} NMOG - NMOG_{bkgd}dt}{\int_{t=0}^{t=end} co - co_{bkgd}dt}$$
Eq. 2

where the excess mixing ratios (ppbv above pre-fire chamber background) of the NMOG and of CO are integrated over the fire from time t=0 to t=end. The emission factors (EF) are in units of gram NMOG emitted per kg dry fuel burned, and are derived from the emission ratios using the carbon mass balance (Akagi et al., 2011; Selimovic et al., 2017):

$$EF_{NMOG} = F_c \times \frac{M_{NMOG}}{M_c} \times \frac{(\Delta NMOG/\Delta CO)}{\sum_{x=1}^n (NC_x \times \frac{\Delta Cx}{\Delta CO})}$$
 Eq. 3

Where  $EF_{NMOG}$  is the emission factor of the NMOG,  $F_c$  is the carbon fraction of the fuel in g/kg,  $M_{NMOG}$  is 543 the molecular mass of the NMOG,  $M_C$  is the molecular mass of carbon, ( $\Delta NMOG/\Delta CO$ ) is the emission ratio of the NMOG relative to CO,  $NC_x$  is the number of carbon in carbon-containing species x and 544 545  $(\Delta C_x/\Delta CO)$  is the emission ratio of species x to CO.  $\Delta$  indicates the excess mixing ratio above background, 546 as is explicitly written in Eq. 2. This method assumes that all of the carbon lost from the fuel as it burns is 547 emitted and measured, which is a reasonable approximation as CO, CO<sub>2</sub>, and CH<sub>4</sub> account for most of the 548 emitted carbon (Akagi et al., 2011). The denominator of the last term estimates total carbon relative to CO. 549 Species C<sub>x</sub> include all species measured by PTR-ToF (excluding overlapped species with FTIR), all species 550 measured by FTIR (including CO, CO<sub>2</sub>, and CH<sub>4</sub>) and black carbon as described by Selimovic et al. (2017). 551 Emission ratios and factors were determined on a fire-by-fire basis, then averaged over all fires (Table 2) 552 or all fires of a particular fuel type (Tables S2 and S3).

The emission ratios and emission factors of the identified compounds averaged over all fires are 554 reported in Table 2. Emission ratios and emission factors of both identified and unidentified compounds 555 for specific fuel types are given in Tables S2 and S3. The large relative standard deviations of both emission 556 ratio and emission factor, for each NMOG, indicate large differences in emission composition between 557 different fires. Analysis of differences in emissions composition between different fuels and combustion 558 processes will be presented in a separate manuscript. Figure 5 compares the average emission ratios 559 determined in this work to several other studies. Our emission ratios have similar values, ranging from a 560 factor of 1.7 higher on average than Gilman et al. (2015) to 0.7 the average of Stockwell et al. (2015). The 561 differences in slopes and scatter are likely due to different fuel types, fire conditions, and sampling 562 strategies. Stockwell et al. (2015) also reported detailed speciation within particular structural categories 563 (non-oxygenated aromatics, phenols, and furans). We compared our speciation for comparable fuel types -564 coniferous canopy, chaparral, and peat - and the agreement for coniferous fuels and chapparal is within a factor of 2 despite differences in ion identification and calibration factor (Fig. 6). The ER to CO are likely 565 566 the easiest way to incorporate this new NMOG data into models since CO emissions from wildfires are 567 relatively well characterized (Liu et al., 2017b).

The 156 PTR ions for which we have identified the NMOG contributors account for a significant 569 fraction of the instrument signal, and total NMOG detected by the PTR-ToF, in each fire. Across all 58 570 stack fires measured with PTR-ToF, an average of 90% of the instrument signal from m/z 12-m/z 217 571 (excluding primary and contaminant ions) is explained by these ions and associated fragments. After 572 calibration, an average of 92% and minimum of 88% of total NMOG mixing ratio detected by PTR-ToF 573 consists of identified compounds (Fig. 7a). The mixing ratios of unidentified species were determined using 574 a calibration factor calculated from the elemental composition of the ion. They are therefore a lower limit 575 and the actual unidentified fraction could be higher (Section 2.3). The PTR-ToF detects about 80-90% of 576 the total NMOG emissions (on a molar basis), based on composition reported by Gilman et al. (2015).

In terms of NMOG mass detected by PTR-ToF, an average of 88% and minimum of 82% is accounted for by identified species (Fig 7b). This is an improvement over Warneke et al. (2011), where 578 579 only 50-75% of the detected mass was identified, and is comparable to Stockwell et al. (2015), with 580 improved identification of emissions from peat, and updated ion assignments (Table S6). Identifying the 581 NMOG contributors to additional ions will not increase this by much, because the remaining (unidentified) 582 ions each account for only a small part of the remaining signal. The unidentified portion is a small fraction 583 of the overall detected emissions, but compared to the identified portion, it consists of species that are 584 heavier, contain more oxygen atoms, and are less volatile (Fig. 8). The average molecular mass of 585 unidentified species is 120 u, compared to 50 u for identified species, and species with 3 or more oxygen 586 atoms comprise 24% of unidentified NMOG emissions, but only 2.5% of identified NMOG emissions. 587 Many of the unidentified emissions are of intermediate volatility, while most identified species are highly 588 volatile. Species that could be efficient SOA precursors may therefore be underrepresented in the list of

identified NMOGs. Additionally, the heavier, more polar unidentified compounds may be preferentially
lost in inlet lines and could comprise a larger fraction of emissions than measured by the PTR-ToF.

The detected and identified NMOGs fall into several broad structural categories: furan-type 592 compounds; benzene-type compounds (aromatics); terpenes; non-aromatic molecules containing oxygen, 593 nitrogen, or sulfur; and other hydrocarbons (mostly alkenes). We also included pyrroles, thiophenes, and 594 pyridines as structural categories, but these account for less than 1% of detected emissions on a molar basis. 595 Terpenes include isoprene, monoterpenes, oxygenated monoterpenes, and sesquiterpenes. Non-aromatic 596 oxygen-containing molecules include alkyl carbonyls, esters, and acids. Non-aromatic nitrogen-containing 597 molecules include HCN, HONO, isocyanic acid, methyl isocyanate, amines (including ammonia), and 598 nitriles. Aromatics and furans include alkyl-substituted and oxygenated derivatives of benzene and furan. 599 On average over all fires, non-aromatic oxygenates were the most abundant, comprising 51% of detected 600 emissions (Fig. 9a). The compounds in each category include a range of functional groups, of which 601 alcohols and carbonyls were the most abundant (Fig. 9b). Many compounds also include an alkene 602 functional group. Some compounds, such as guaiacol, have several functional groups. In these cases, the 603 NMOG was counted once in each category.

Compared to several previous laboratory studies reporting highly chemically detailed emissions 605 using GC instruments (Hatch et al., 2015;Gilman et al., 2015;Hatch et al., 2017), we observed a similar 606 range and type of speciation for non-oxygenated aromatics, thiophenes, pyrroles, pyridines, alkyl nitriles, 607 alkyl ketones, alkyl esters, and small alcohols. However, this work and a previous PTR-MS study 608 (Stockwell et al., 2015) also observed more highly substituted oxygen-containing aromatics and furans, 609 such as hydroxymethylfuranone and syringol. These substituted compounds contribute significant 610 additional reactivity. For example, Gilman et al. (2015), who studied similar fuels, reported OH reactivity of 1.3-5.5 s<sup>-1</sup> (ppm CO)<sup>-1</sup> for furans. In this study, the average OH reactivity of furans is 14.2 s<sup>-1</sup> (ppm CO)<sup>-1</sup> 611 <sup>1</sup>. The SOA yields of many of these compounds are unknown but they are likely important SOA precursors 612 613 (Yee et al., 2013;Gilman et al., 2015;Hatch et al., 2017;Bruns et al., 2016).

Reaction with the hydroxyl radical (•OH) is an important removal pathway for gas-phase biomass 615 burning emissions in the atmosphere. NMOGs have been previously shown to be an important sink for the 616 OH radical, despite comprising less than 1% by mass of the total measured gas-phase emissions (Gilman 617 et al., 2015). We compiled the rate constants with •OH of the identified species. Where an experimentally 618 determined rate constant was not available, the rate constant of a structurally similar species was used (rate 619 constants and citations in Table S5). On average, furans, aromatics, terpenes, and non-aromatic oxygenates 620 contribute a roughly equal amount to total OH reactivity (Fig. 10a). It has been shown that the average 621 reactivity of NMOG emissions can vary greatly between fuel types (Gilman et al., 2015); here, we show 622 that the average reactivity, and the types of compounds that contribute most to reactivity, also vary greatly

over the course of a fire (Fig. 10b). The spike in average reactivity at the beginning of the fire is due todistillation of terpenes.

The volatility distribution of emitted species also changes over the course of these lab fires. We 626 determined the saturation vapor concentration (C<sub>0</sub>, in µg m<sup>-3</sup> at 25°C) for each of the identified and 627 unidentified species. The values were taken from databases (CRC Handbook, NIST Chemistry WebBook, 628 (Yaws, 2015)) or estimated based on elemental composition via the parameterization described by Li et al. 629 (2016). Species emitted from lower temperature processes during the fire have a higher fraction of 630 compounds with low volatility compared to the high-temperature processes (later and earlier in the fire 631 shown in Fig. 11). Further discussion of chemical differences, and low- and high-temperature processes, 632 will be presented in a separate manuscript (Sekimoto et al., 2018). The PTR-ToF measures mostly species whose volatility is classified as volatile organic compounds (VOC,  $C_0 > 3 \times 10^6 \ \mu g \ m^{-3}$ ), and a few 633 intermediate volatility compounds (IVOC,  $300 < C_0 < 3 \times 10^6 \,\mu g \,m^{-3}$ ) and semivolatile compounds (SVOC, 634  $0.3 < C_0 < 300 \ \mu g \ m^{-3}$ ) are detected. Many more IVOC species have been measured by 2D-GC (Hatch et 635 al., 2017). It is expected that many species of  $C_0 < 10^4 \,\mu g \, m^{-3}$  were not transmitted through the transfer inlet 636 637 and instrument tubing quickly enough to be quantifiable by the PTR-MS (Pagonis et al., 2017).

#### 638 4. Conclusions

Gas-phase emissions of NMOGs and some inorganic compounds were measured with a high-640 resolution PTR-ToF instrument during the FIREX 2016 laboratory intensive. Using a combination of 641 techniques, including GC pre-separation, NO<sup>+</sup> CIMS, and time-series correlation, we have identified many 642 more compounds and with greater certainty than has been reported in previous PTR-MS studies of biomass 643 burning emissions. We have identified the NMOG contributors to ~90% of the PTR-ToF signal, accounting 644 for ~90% of the NMOG mass detected by the instrument, and determined the emission factors of these compounds. The NMOG ions not identified are in general larger, more oxygenated, and less volatile than 645 646 the identified species. This should be considered if using PTR-ToF to study SOA precursors. Unidentified 647 compounds may also be preferentially lost in inlets. The PTR-ToF measurement generally agrees well with 648 other instrumentation for many species. However, small, multiply-oxygenated species such as glyoxal and 649 methylglyoxal may have significant interferences. We determined the reaction rate constant of each 650 identified NMOG with the OH radical. Furans, aromatics, and terpenes are the most important reactive 651 species measured by PTR-ToF. We show that the reactivity of the emissions, volatility of the emissions, 652 and the compounds that contribute to the reactivity can change considerably as different combustion 653 processes occur.

This work provides a guide to interpreting PTR-ToF measurements of biomass burning that is strongly supported by literature and complementary analytical techniques. This will serve as a foundation for future use of FIREX 2016 PTR-ToF data, and interpretation of PTR-ToF field measurements. Finally,

- this work provides the best available emission factors and emission ratios to CO for many wildfire-
- generated NMOGs.

#### 659 **5. Acknowledgements**

- 660 A. Koss acknowledges funding from the NSF Graduate Fellowship Program. K. Sekimoto acknowledges
- funding from the Postdoctoral Fellowships for Research Abroad from Japan Society for the Promotion of
- Science (JSPS) and a Grant-in-Aid for Young Scientists (B) (15K16117) from the Ministry of Education,
- Culture, Sports, Science and Technology of Japan. R. Yokelson and V. Selimovic were supported by
- NOAA-CPO grant NA16OAR4310100. J. R. Krechmer and J. L. Jimenez were supported by DOE
- (BER/ASR) DE-SC0016559. We thank the USFS Missoula Fire Sciences Laboratory for their help in
- conducting these experiments. This work was also supported by NOAA's climate Research and Health of
- the Atmosphere initiative.

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

# Table 1

Instrumentation details.

| Instrument                | Operating principle                                                                                           | Species measured                                               | Time<br>resolution                                   | Detection limits                                                                 | Inlet setup                                                                                                                                                                                                                             | Reference                                                 |
|---------------------------|---------------------------------------------------------------------------------------------------------------|----------------------------------------------------------------|------------------------------------------------------|----------------------------------------------------------------------------------|-----------------------------------------------------------------------------------------------------------------------------------------------------------------------------------------------------------------------------------------|-----------------------------------------------------------|
| PTR-ToF                   | Chemical ionization mass<br>spectrometry; H <sub>3</sub> O <sup>+</sup><br>reagent ions                       | Polar and unsaturated<br>NMOG (several<br>hundred)             | 2 Hz                                                 | 20 pptv<br>(acrylonitrile) to<br>2.6 ppb (H <sub>2</sub> S)<br>at 1Hz resolution | Stack: from sampling platform, 16m long.<br>Room: from 3m above combustion<br>chamber floor, 7m long.<br>Both: <sup>1</sup> / <sub>2</sub> " OD PFA inlet, 40°C, flow rate<br>100 lpm. Subsample 500 sccm through<br>PEEK capillary.    | Yuan et al.<br>(2016)                                     |
| NO <sup>+</sup> -<br>CIMS | Chemical ionization mass<br>spectrometry; NO <sup>+</sup><br>reagent ions                                     | Saturated, unsaturated,<br>and polar NMOG<br>(several hundred) | 2 Hz                                                 | 20 pptv<br>(aromatics) to 19<br>ppb (methanol)<br>at 1Hz resolution              | Same as PTR-ToF.                                                                                                                                                                                                                        | Koss et al.,<br>(2016)                                    |
| GC-EI-<br>MS              | Gas chromatographic<br>(GC) separation with<br>electron-ionization<br>quadrupole mass<br>spectrometry (EI-MS) | NMOG (several<br>hundred)                                      | 4 minute<br>sample (240<br>sccm) every<br>20 minutes | <5 pptv (most<br>species)<br>for 4-minute<br>sample                              | Stack: from sampling platform, 16m long,<br>1/2" OD PFA inlet, flow rate 20 lpm.<br>Room: from 3m above combustion<br>chamber floor, 7m long, 1/4" OD PFA, flow<br>rate 2-7 lpm.<br>Both: Dynamically diluted with UHP N <sub>2</sub> . | Lerner et al. (2017)                                      |
| GC-CIMS                   | Gas chromatographic<br>separation with chemical<br>ionization mass<br>spectrometry (CIMS)                     | Polar and unsaturated<br>NMOG (several<br>hundred)             | 4 minute<br>sample<br>every 20<br>minutes            | qualitative<br>measurement only                                                  | Same as GC-EI-MS                                                                                                                                                                                                                        | (this work)                                               |
| OP-FTIR                   | Open path FTIR<br>absorption spectroscopy                                                                     | Small organic and<br>inorganic trace gases<br>(about 20)       | 0.73 Hz                                              | 1 ppbv<br>at 0.73 Hz<br>resolution                                               | From sampling platform (no inlet).                                                                                                                                                                                                      | Stockwell et<br>al. (2014);<br>Selimovic et<br>al. (2017) |
| ACES                      | Broadband cavity<br>enhanced spectroscopy<br>("Airborne Cavity<br>Enhanced Spectrometer"                      | Glyoxal, NO <sub>2</sub> , HONO,<br>methyl glyoxal             | 1 Hz                                                 | 100 pptv (glyoxal)<br>to 2 ppbv<br>(HONO); ~5 ppbv<br>for methylgyloxal          | Stack: from sampling platform, 1m long<br>1/4" OD PFA including particle filter                                                                                                                                                         | Min et al.<br>(2016)                                      |
| I <sup>-</sup> CIMS       | Chemical ionization mass<br>spectrometry; I <sup>-</sup> reagent<br>ions                                      | Polar NMOG (several hundred)                                   | 1 Hz                                                 | 1 pptv (malonic<br>acid) to 1.5 ppbv<br>(peroxyacetic acid)                      | Shared with PTR-ToF.<br>Stack: from sampling platform, 16m long.                                                                                                                                                                        | Lee et al. (2014)                                         |

|  |  | at 1Hz resolution | Room: from 3m above combustion                                  |  |
|--|--|-------------------|-----------------------------------------------------------------|--|
|  |  |                   | chamber floor, 7m long.                                         |  |
|  |  |                   | Both: <sup>1</sup> / <sub>2</sub> " OD PFA inlet, flow rate 100 |  |
|  |  |                   | lpm.                                                            |  |

#### Table 2

Ion exact masses, formulas, and NMOG contributor(s); and the emission ratios and emission factors of those contributors.

| Ion exact |                                                | NMOG contributor(s)                     | ER to CO,       |                       |
|-----------|------------------------------------------------|-----------------------------------------|-----------------|-----------------------|
| m/z (Th)  | Ion Formula                                    | (details in Table S1)                   | ppb/ppm (σ)     | EF, g/kg ( $\sigma$ ) |
| 18.034    | NH <sub>3</sub> H <sup>+</sup>                 | ammonia                                 | 17 (13)         | 0.82 (0.80)           |
| 26.015    | $C_2H_2^+$                                     | acetylene                               | 5.0 (2.5)       | 0.36 (0.24)           |
| 28.018    | HCNH <sup>+</sup>                              | Hydrogen cyanide                        | 3.9 (3.6)       | 0.33 (0.47)           |
| 28.031    | $C_2H_4^+$                                     | ethene                                  | 7.1 (3.8)       | 0.54 (0.38)           |
| 30.034    | CH <sub>3</sub> NH <sup>+</sup>                | Methanimine                             | 0.0092 (0.012)  | 0.00073 (0.0010)      |
| 31.018    | CH <sub>2</sub> OH <sup>+</sup>                | Formaldehyde                            | 20 (10)         | 1.7 (1.2)             |
| 33.034    | $CH_4OH^+$                                     | Methanol                                | 12 (5.9)        | 1.1 (0.82)            |
| 34.995    | $H_2SH^+$                                      | Hydrogen sulfide                        | 0.26 (0.51)     | 0.029 (0.062)         |
| 42.034    | $C_2H_3NH^+$                                   | Acetonitrile                            | 1.0 (1.4)       | 0.13 (0.22)           |
| 43.054    | $C_3H_6H^+$                                    | Propene                                 | 4.5 (2.9)       | 0.55 (0.44)           |
| 44.013    | HNCOH <sup>+</sup>                             | Isocyanic acid                          | 4.6 (2.5)       | 0.53 (0.34)           |
| 44.050    | $C_2H_5NH^+$                                   | Etheneamine                             | 0.052 (0.055)   | 0.0064 (0.0069)       |
| 45.034    | $C_2H_4OH^+$                                   | Acetaldehyde                            | 7.4 (5.2)       | 0.92 (0.73)           |
| 46.029    | CH <sub>3</sub> NOH <sup>+</sup>               | Formamide                               | 0.10 (0.12)     | 0.013 (0.018)         |
| 46.065    | C <sub>2</sub> H <sub>7</sub> NH <sup>+</sup>  | Ethylamine                              | 0.0030 (0.0080) | 0.00038 (0.0010)      |
| 47.013    | $CH_2O_2H^+$                                   | Formic acid                             | 2.2 (1.4)       | 0.28 (0.22)           |
| 47.049    | C <sub>2</sub> H <sub>6</sub> OH <sup>+</sup>  | ethanol                                 | 0.56 (0.92)     | 0.072 (0.11)          |
| 48.008    | HNO <sub>2</sub> H <sup>+</sup>                | Nitrous acid                            | 4.1 (1.8)       | 0.49 (0.23)           |
| 49.011    | CH <sub>4</sub> SH <sup>+</sup>                | methane thiol                           | 0.13 (0.27)     | 0.020 (0.043)         |
| 49.028    | $CH_4O_2H^+$                                   | Methanediol                             | 0.0040 (0.0028) | 0.00051 (0.00039)     |
| 52.018    | C <sub>3</sub> HNH <sup>+</sup>                | Propyne nitrile                         | 0.0090 (0.0068) | 0.0013 (0.0011)       |
| 53.039    | $C_4H_4H^+$                                    | 1-Buten-3-yne                           | 0.35 (0.20)     | 0.049 (0.035)         |
| 54.034    | C <sub>3</sub> H <sub>3</sub> NH <sup>+</sup>  | acrylonitrile                           | 0.16 (0.12)     | 0.025 (0.021)         |
| 55.018    | $C_3H_2OH^+$                                   | 2-propynal                              | 0.20 (0.10)     | 0.029 (0.019)         |
| 55.054    | $C_4H_6H^+$                                    | Butadienes                              | 1.8 (1.2)       | 0.28 (0.23)           |
| 56.050    | C <sub>3</sub> H <sub>5</sub> NH <sup>+</sup>  | Propanenitrile                          | 0.10 (0.14)     | 0.017 (0.027)         |
| 57.034    | C <sub>3</sub> H <sub>4</sub> OH <sup>+</sup>  | Acrolein                                | 5.4 (3.0)       | 0.80 (0.52)           |
| 57.070    | $C_4H_8H^+$                                    | Butenes, other hydrocarbon              | 1.2 (1.0)       | 0.21 (0.21)           |
| 58.029    | C <sub>2</sub> H <sub>3</sub> NOH <sup>+</sup> | methyl isocyanate, hydroxy acetonitrile | 0.089 (0.086)   | 0.015 (0.016)         |
| 58.065    | C <sub>3</sub> H <sub>7</sub> NH <sup>+</sup>  | Propene amine                           | 0.022 (0.034)   | 0.0036 (0.0059)       |
| 59.013    | $C_2H_2O_2H^+$                                 | glyoxal                                 | 1.7 (1.3)       | 0.26 (0.23)           |
| 59.049    | $C_3H_6OH^+$                                   | Acetone                                 | 2.3 (1.7)       | 0.39 (0.35)           |
| 60.044    | C <sub>2</sub> H <sub>5</sub> NOH <sup>+</sup> | acetamide                               | 0.46 (1.1)      | 0.086 (0.21)          |
| 60.081    | C <sub>3</sub> H <sub>9</sub> NH <sup>+</sup>  | C3 amines                               | 0.023 (0.052)   | 0.0041 (0.010)        |
| 61.028    | $C_2H_4O_2H^+$                                 | acetic acid, glycolaldehyde             | 15 (11)         | 2.5 (2.2)             |
| 62.024    | CH <sub>3</sub> NO <sub>2</sub> H <sup>+</sup> | nitromethane                            | 0.34 (0.21)     | 0.053 (0.036)         |
| 63.026    | $C_2H_6SH^+$                                   | Dimethyl sulfide                        | 0.012 (0.018)   | 0.0024 (0.0041)       |
| 66.034    | C <sub>4</sub> H <sub>3</sub> NH <sup>+</sup>  | butynenitriles, cyanoallene             | 0.0020 (0.0017) | 0.00037 (0.00035)     |
| 67.054    | $C_5H_6H^+$                                    | 1,3-cyclopentadiene                     | 0.16 (0.13)     | 0.030 (0.029)         |
| 68.050    | C <sub>4</sub> H <sub>5</sub> NH <sup>+</sup>  | butenenitrile isomers, pyrrole          | 0.36 (0.46)     | 0.071 (0.10)          |
| 68.997    | $C_3O_2H^+$                                    | carbon suboxide                         | 0.016 (0.0093)  | 0.0028 (0.0018)       |
| 69.034    | $C_4H_4OH^+$                                   | furan                                   | 1.9 (1.1)       | 0.36 (0.25)           |
| 69.070    | $C_5H_8H^+$                                    | Isoprene                                | 1.0 (0.82)      | 0.21 (0.20)           |
| 70.065    | $C_4H_7NH^+$                                   | Butanenitriles, dihydropyrrole          | 0.076 (0.12)    | 0.016 (0.028)         |

| 71.013  | $C_3H_2O_2H^+$                                 | Propiolic acid                           | 0.046 (0.025)   | 0.0088 (0.0057)   |
|---------|------------------------------------------------|------------------------------------------|-----------------|-------------------|
| 71.049  | $C_4H_6OH^+$                                   | MVK, methacrolein, crotonaldehyde        | 1.7 (1.0)       | 0.32 (0.21)       |
| 71.086  | $C_5H_{10}H^+$                                 | Pentenes, methylbutenes                  | 0.12 (0.11)     | 0.026 (0.029)     |
| 72.081  | $C_4H_9NH^+$                                   | butene amines, tetrahydropyrrole         | 0.0077 (0.014)  | 0.0016 (0.0031)   |
| 73.028  | $C_3H_4O_2H^+$                                 | methyl glyoxal, acrylic acid             | 1.4 (1.0)       | 0.28 (0.19)       |
| 73.065  | $C_4H_8OH^+$                                   | MEK, 2-methylpropanal, butanal           | 0.52 (0.50)     | 0.11 (0.13)       |
| 74.024  | $C_2H_3NO_2H^+$                                | nitroethene                              | 0.0068 (0.0039) | 0.0013 (0.00084)  |
| 75.044  | C U O U                                        | hydroxyacetone, methyl acetate, ethyl    | 2.8 (2.3)       | 0.55 (0.45)       |
| /5.044  | $C_3H_6O_2H^2$                                 | formate                                  |                 | ``´´              |
| 76.039  | $C_2H_5NO_2H^+$                                | nitroethane                              | 0.0034 (0.0022) | 0.00072 (0.00057) |
| 79.001  |                                                | n-sulfinylmethanamine                    | 0.00031         | 6.9e-05 (5.9e-05) |
| /8.001  | CH <sub>3</sub> NOSH                           |                                          | (0.00022)       |                   |
| 79.054  | $C_6H_6H^+$                                    | benzene                                  | 1.7 (1.1)       | 0.37 (0.30)       |
| 80.050  | $C_5H_5NH^+$                                   | pyridine, C5 nitriles                    | 0.13 (0.18)     | 0.031 (0.049)     |
| 81.024  |                                                | 2,4-Cyclopentadiene-1-one, other         | 0.61 (0.40)     | 0.13 (0.093)      |
| 81.034  | C5H4OH                                         | hydrocarbon                              |                 |                   |
| 82.065  | $C_5H_7NH^+$                                   | methylpyrrole, pentenenitriles           | 0.093 (0.15)    | 0.023 (0.041)     |
| 83.049  | $C_5H_6OH^+$                                   | methylfurans, other hydrocarbon          | 1.51 (1.01)     | 0.35 (0.28)       |
| 84.081  | $C_5H_9NH^+$                                   | Pentanenitriles                          | 0.035 (0.066)   | 0.0094 (0.019)    |
| 85.011  | $C_4H_4SH^+$                                   | thiophene                                | 0.057 (0.041)   | 0.014 (0.011)     |
| 85.028  | $C_4H_4O_2H^+$                                 | 2-(3H)-furanone                          | 1.7 (1.1)       | 0.39 (0.30)       |
| 95.065  |                                                | 3-methyl-3-butene-2-one, cyclopentanone, | 0.52 (0.35)     | 0.12 (0.10)       |
| 83.003  | C5H8OH                                         | other hydrocarbon                        |                 |                   |
| 97.044  |                                                | 2,3-butanedione, methyl acrylate, other  | 2.0 (1.7)       | 0.46 (0.35)       |
| 87.044  | $C_4\Pi_6O_2\Pi$                               | hydrocarbon                              |                 |                   |
| 87.080  |                                                | 3-methyl-2-butanone, methylbutanals,     | 0.16 (0.20)     | 0.042 (0.059)     |
| 87.080  | C5H10OH                                        | pentanones                               |                 |                   |
| 89.023  | $C_3H_4O_3H^+$                                 | pyruvic acid                             | 0.041 (0.027)   | 0.010 (0.0070)    |
| 89.060  | $C_4H_8O_2H^+$                                 | Methyl propanoate                        | 0.34 (0.27)     | 0.081 (0.067)     |
| 90.055  | $C_3H_7NO_2H^+$                                | nitropropanes                            | 0.0022 (0.0037) | 0.00056 (0.0010)  |
| 92.050  | $C_6H_6N^+$                                    | ethynylpyrrole                           | 0.0066 (0.0054) | 0.0017 (0.0016)   |
| 93.070  | $C_7H_8H^+$                                    | toluene                                  | 0.9 (0.72)      | 0.24 (0.24)       |
| 94.029  | C <sub>5</sub> H3NOH <sup>+</sup>              | Furan carbonitriles                      | 0.01 (0.012)    | 0.0031 (0.0044)   |
| 94.065  | $C_6H_7NH^+$                                   | methylpyridines                          | 0.075 (0.12)    | 0.022 (0.037)     |
| 94.998  | $C_2H_6S_2H^+$                                 | dimethyl disulfide                       | 0.0082 (0.0064) | 0.0022 (0.0020)   |
| 95.049  | $C_6H_6OH^+$                                   | phenol                                   | 2.0 (1.4)       | 0.55 (0.46)       |
| 96.044  | C <sub>5</sub> H <sub>5</sub> NOH <sup>+</sup> | 4-Pyridinol                              | 0.048 (0.071)   | 0.014 (0.021)     |
| 96.081  | $C_6H_9NH^+$                                   | C2-substituted pyrroles                  | 0.043 (0.081)   | 0.013 (0.025)     |
| 97.028  | $C_5H_4O_2H^+$                                 | Furfurals, other hydrocarbons            | 2.1 (1.4)       | 0.60 (0.58)       |
| 97.065  | C <sub>6</sub> H <sub>8</sub> OH <sup>+</sup>  | C2-substituted furans                    | 0.83 (0.65)     | 0.22 (0.20)       |
| 98.096  | $C_6H_{11}NH^+$                                | 4-methylpentanenitrile                   | 0.013 (0.026)   | 0.004 (0.0084)    |
| 99.026  | $C_5H_6SH^+$                                   | methylthiophene                          | 0.079 (0.072)   | 0.021 (0.020)     |
| 99.044  | $C_5H_6O_2H^+$                                 | 2-methanol furanone                      | 1.5 (1.1)       | 0.40 (0.31)       |
| 99.080  | $C_6H_{10}OH^+$                                | Methylcyclopentanone, cyclohexanone,     | 0.086 (0.087)   | 0.024 (0.028)     |
|         |                                                | hexenones                                |                 |                   |
| 101.023 | $C_4H_4O_3H^+$                                 | Dihydrofurandione                        | 0.18 (0.15)     | 0.052 (0.052)     |
| 101.060 | $C_5H_8O_2H^+$                                 | methyl methacrylate, other hydrocarbon   | 0.51 (0.34)     | 0.14 (0.10)       |
| 101.096 | $C_6H_{12}OH^+$                                | hexanals, hexanones                      | 0.017 (0.021)   | 0.0052 (0.0072)   |
| 103.039 | $C_4H_6O_3\overline{H^+}$                      | acetic anhydride,                        | 0.34 (0.28)     | 0.092 (0.075)     |
| 103.054 | $C_8H_6H^+$                                    | Phenylacetylene                          | 0.039 (0.037)   | 0.011 (0.012)     |
| 104.049 | $C_7H_5NH^+$                                   | Benzonitrile                             | 0.076 (0.057)   | 0.023 (0.024)     |
| 105.070 | $C_8H_8H^+$                                    | styrene                                  | 0.27 (0.21)     | 0.079 (0.073)     |
| 106.065 | $C_7H_7NH^+$                                   | vinyl pyridine                           | 0.010 (0.011)   | 0.0033 (0.0038)   |
| 107.049 | $C_7H_6OH^+$                                   | benzaldehyde                             | 0.26 (0.15)     | 0.079 (0.056)     |

| 107.086 | $C_8H_{10}H^+$                                 | C8 aromatics                                                               | 0.40 (0.33)     | 0.13 (0.13)      |
|---------|------------------------------------------------|----------------------------------------------------------------------------|-----------------|------------------|
| 108.044 | C <sub>6</sub> H <sub>5</sub> NOH <sup>+</sup> | pyridine aldehyde                                                          | 0.018 (0.015)   | 0.0058 (0.0059)  |
| 108.081 | $C_7H_9NH^+$                                   | dimethyl + ethyl pyridine, heptyl nitriles                                 | 0.027 (0.052)   | 0.009 (0.018)    |
| 109.028 | $C_6H_4O_2H^+$                                 | Quinone                                                                    | 0.34 (0.27)     | 0.093 (0.065)    |
| 109.065 | $C_7H_8OH^+$                                   | Cresol, anisole                                                            | 1.5 (1.0)       | 0.46 (0.39)      |
| 110.096 | $C_7H_{11}NH^+$                                | C7 acrylonitriles, C3-substituted pyrroles                                 | 0.017 (0.032)   | 0.0057 (0.012)   |
| 111.044 | $C_6H_6O_2H^+$                                 | methyl furfural, benzene diols, 2-acetyl furan                             | 2.4 (1.4)       | 0.75 (0.62)      |
| 111.080 | $C_7H_{10}OH^+$                                | C3-substituted furans, other compounds                                     | 0.3 (0.27)      | 0.093 (0.10)     |
| 112.039 | $C_5H_5NO_2H^+$                                | dihydroxy pyridine, methyl maleimide                                       | 0.021 (0.023)   | 0.0071 (0.0088)  |
| 113.023 | $C_5H_4O_3H^+$                                 | 5-Hydroxy 2-furfural, 2-furoic acid                                        | 0.32 (0.22)     | 0.11 (0.10)      |
| 113.060 | $C_6H_8O_2H^+$                                 | 2-hydroxy-3-methyl-2-cyclopenten-1-one                                     | 0.67 (0.50)     | 0.21 (0.17)      |
| 113.096 | $C_7H_{12}OH^+$                                | ethyl cyclopentanone                                                       | 0.036 (0.034)   | 0.012 (0.013)    |
| 114.019 | $C_4H_3NO_3H^+$                                | nitrofuran                                                                 | 0.0037 (0.0025) | 0.0012 (0.001)   |
| 115.039 | $C_5H_6O_3H^+$                                 | 5-hydroxymethyl-2[3H]-furanone                                             | 0.63 (0.52)     | 0.20 (0.18)      |
| 115.075 | $C_6H_{10}O_2H^+$                              | C6 diketone isomers, C6 esters                                             | 0.10 (0.074)    | 0.032 (0.028)    |
| 115.112 | $C_7H_{14}OH^+$                                | Heptanal, 2,4-dimethyl-3-pentanone, heptanone                              | 0.030 (0.030)   | 0.010 (0.011)    |
| 117.055 | $C_5H_8O_3H^+$                                 | 5-hydroxymethyl tetrahydro 2-furanone, 5-<br>hydroxy tetrahydro 2-furfural | 0.43 (0.50)     | 0.13 (0.12)      |
| 117.070 | $C_9H_8H^+$                                    | Indene, methyl ethynyl benzene                                             | 0.081 (0.081)   | 0.027 (0.031)    |
| 117.091 | $C_6H_{12}O_2H^+$                              | butyl ester acetic acid, other C6 esters                                   | 0.033 (0.045)   | 0.012 (0.019)    |
| 118.050 | $C_4H_7NO_3H^+$                                | butene nitrates                                                            | 0.008 (0.0066)  | 0.0027 (0.0025)  |
| 118.065 | $C_8H_7NH^+$                                   | Benzeneacetonitrile                                                        | 0.032 (0.039)   | 0.011 (0.015)    |
| 119.049 | $C_8H_6OH^+$                                   | benzofuran                                                                 | 0.12 (0.088)    | 0.038 (0.029)    |
| 119.086 | $C_9H_{10}H^+$                                 | MethylStyrene, propenyl benzene+methyl ethenyl benzene, indane             | 0.12 (0.10)     | 0.043 (0.043)    |
| 120.081 | C <sub>8</sub> H <sub>9</sub> NH <sup>+</sup>  | dihydro pyridine                                                           | 0.0049 (0.0075) | 0.0018 (0.0029)  |
| 121.065 | $C_8H_8OH^+$                                   | Tolualdehyde                                                               | 0.34 (0.31)     | 0.11 (0.11)      |
| 121.101 | $C_9H_{12}H^+$                                 | C9 aromatics                                                               | 0.15 (0.13)     | 0.056 (0.060)    |
| 123.044 | $C_7H_6O_2H^+$                                 | Salicyladehyde                                                             | 0.21 (0.15)     | 0.074 (0.070)    |
| 123.080 | $C_8H_{10}OH^+$                                | ethylphenol+dimethylphenol, methylanisole                                  | 0.37 (0.28)     | 0.13 (0.12)      |
| 124.039 | $C_6H_5NO_2H^+$                                | nitrobenzene                                                               | 0.019 (0.013)   | 0.0068 (0.0062)  |
| 125.023 | $C_6H_4O_3H^+$                                 | hydroxy benzoquinone                                                       | 0.18 (0.10)     | 0.060 (0.044)    |
| 125.060 | $C_7H_8O_2H^+$                                 | guaiacol                                                                   | 1.3 (1.0)       | 0.48 (0.59)      |
| 126.128 | $C_8H_{15}NH^+$                                | C8 nitriles                                                                | 0.0015 (0.0042) | 0.00062 (0.0017) |
| 126.970 | $C_2H_6S_3H^+$                                 | dimethyl trisulfide                                                        | 0.0024 (0.0036) | 0.00081 (0.0011) |
| 127.039 | $C_6H_6O_3H^+$                                 | 5-hydroxymethyl 2-furfural                                                 | 0.88 (0.65)     | 0.32 (0.32)      |
| 129.055 | $C_6H_8O_3H^+$                                 | 2,5-di(hydroxymethyl)furan, methyl hydroxy dihydrofurfural                 | 0.39 (0.27)     | 0.14 (0.13)      |
| 129.070 | $C_{10}H_8H^+$                                 | Naphthalene                                                                | 0.20 (0.16)     | 0.07 (0.067)     |
| 131.086 | $C_{10}H_{10}H^+$                              | Dihydronaphthalene                                                         | 0.078 (0.063)   | 0.030 (0.030)    |
| 132.081 | C <sub>9</sub> H <sub>9</sub> NH <sup>+</sup>  | MethylBenzeneAcetonitrile                                                  | 0.014 (0.020)   | 0.0056 (0.0088)  |
| 133.065 | C <sub>9</sub> H <sub>8</sub> OH <sup>+</sup>  | Methylbenzofurans                                                          | 0.19 (0.35)     | 0.068 (0.11)     |
| 133.101 | $C_{10}H_{12}H^+$                              | EthylStyrene, butenyl benzene isomers,<br>MethylIndane                     | 0.086 (0.071)   | 0.034 (0.033)    |
| 135.080 | $C_9H_{10}OH^+$                                | methyl acetophenone                                                        | 0.11 (0.073)    | 0.041 (0.033)    |
| 135.117 | $C_{10}H_{14}H^+$                              | C10 aromatics                                                              | 0.11 (0.10)     | 0.045 (0.049)    |
| 137.060 | $C_8H_8O_2H^+$                                 | Methylbenzoicacid                                                          | 0.22 (0.13)     | 0.083 (0.063)    |
| 137.132 | $C_{10}H_{16}H^+$                              | monoterpenes                                                               | 2.7 (4.2)       | 1.1 (2.0)        |
| 138.055 | $C_7H_7NO_2H^+$                                | nitrotoluene                                                               | 0.019 (0.023)   | 0.0080 (0.011)   |
| 139.075 | $C_8H_{10}O_2H^+$                              | methylguiacol                                                              | 0.77 (0.63)     | 0.34 (0.46)      |
| 143.086 | $C_{11}H_{10}H^+$                              | Methyl naphthalene                                                         | 0.08 (0.063)    | 0.033 (0.032)    |
| 145.050 | $C_6H_8O_4H^+$                                 | Levoglucosan pyrolysis product                                             | 0.35 (0.27)     | 0.15 (0.17)      |

| 145.065 | $C_{10}H_8OH^+$      | 2-ethenyl benzofuran                   | 0.05 (0.037)    | 0.020 (0.018)   |
|---------|----------------------|----------------------------------------|-----------------|-----------------|
| 145.101 | $C_{11}H_{12}H^+$    | ethylindene                            | 0.037 (0.036)   | 0.016 (0.019)   |
| 147.080 | $C_{10}H_{10}OH^+$   | dimethylbenzofuran, ethyl benzofuran   | 0.10 (0.065)    | 0.043 (0.034)   |
| 149.096 | $C_{10}H_{12}OH^+$   | estragole                              | 0.069 (0.066)   | 0.029 (0.033)   |
| 149.132 | $C_{11}H1_6H^+$      | C11 aromatics                          | 0.026 (0.022)   | 0.012 (0.012)   |
| 151.075 | $C_9H_{10}O_2H^+$    | vinylguaiacol                          | 0.35 (0.31)     | 0.15 (0.16)     |
| 153.055 | $C_8H_8O_3H^+$       | Vanillin                               | 0.37 (0.31)     | 0.17 (0.22)     |
| 153.070 | $C_{12}H_8H^+$       | acenaphthylene                         | 0.025 (0.026)   | 0.010 (0.013)   |
| 153.127 | $C_{10}H_{16}OH^+$   | Camphor, other oxygenated monoterpenes | 0.070 (0.15)    | 0.031 (0.066)   |
| 155.070 | $C_8H_{10}O_3H^+$    | syringol                               | 0.12 (0.14)     | 0.046 (0.055)   |
| 155.143 | $C_10H_{18}OH^+$     | Cineole, other oxygenated monoterpenes | 0.013 (0.012)   | 0.0059 (0.0061) |
| 157.101 | $C_{12}H_{12}H^+$    | C2-substituted naphthalenes            | 0.051 (0.039)   | 0.024 (0.025)   |
| 157.159 | $C_{10}H_{20}OH^+$   | Decanal                                | 0.0051 (0.0051) | 0.0024 (0.0030) |
| 163.148 | $C_{12}H_{18}H^+$    | C12 aromatics                          | 0.013 (0.012)   | 0.0067 (0.0073) |
| 165.091 | $C_{10}H_{12}O_2H^+$ | Eugenol, isoeugenol                    | 0.22 (0.17)     | 0.11 (0.12)     |
| 177.164 | $C_{13}H_{20}H^+$    | C13 aromatics                          | 0.0094 (0.0079) | 0.0053 (0.0058) |
| 205.195 | $C_{15}H_{24}H^+$    | Sesquiterpenes                         | 0.15 (0.13)     | 0.090 (0.090)   |

Comparison of measured and calculated calibration factors for several NMOGs. The nine compounds used to determine the calibration proportionality constant are highlighted as red squares. The shaded area shows an uncertainty of +10%/-50%. HONO, HCN, and ammonia sensitivities were derived from comparison with FTIR and are included as "measured" sensitivities.