# Peer review of "Non-methane organic gas emissions from biomass burning"

_Atmospheric Chemistry and Physics, 2017_

## Referee Comment (RC1) · Anonymous Referee #1 · 8 Dec 2017

This paper describes the identification of non-methane organic gases (NMOGs) measured by PTR-ToF mass spectrometry of biomass burning emissions during the 2016 FIREX experiment at the Missoula Fire lab. The emissions of the identified species are quantified and emission ratios and factors for the different species are derived. The measurements for individual organic gases are also compared with simultaneous measurements using other techniques, such as chemical ionization mass spectrometry (using I- and NO+) and infrared and UV spectroscopy. The authors find that while non-aromatic oxygenated compounds are the most abundant of the identified species,

furans and aromatics comprise a large portion f the OH reactivity of the fresh emissions.

This is a well-executed study on an important topic in atmospheric chemistry, as the amount of O3 and SOA attributable to fires is still highly uncertain, in part because of uncertainties in the emissions of NMOGs. In addition, the paper and the supplemental data go into a high level of detail on the measurements, calibrations, and subsequent calculations, making it very clear what was done, what assumptions were made, and the level of certainty for each of the provided values. The manuscript was also remarkably free of typos and other errors. I recommend publication after very minor revisions to address the few issues I noticed, listed below.

L183: What does the acronym "PLOT" mean here?

L185: Who was the manufacturer of the "MXT-624" column? I'm assuming that's a catalogue number for some specific manufacturer?

L327: How does the disagreement in identification discussed here potentially affect current review values of biomass burning emission factors?

L327 and elsewhere: Please refer to the tab of Table S1 you are referring to, or make each tab of the Excel spreadsheet it's own table number (e.g., Table S1, S2, ... or Table S1a, S1b, ...)

L420: This reference to Table 1 is probably redundant at this point.

L445-447: My understanding is that the FTIR uncertainty in HONO is also very high, around 10 ppbv. Is this correct, and if so, does that affect the use of the FTIR to determine the PRT-ToF level of detection?

L457-459: Can you discuss how the slopes and correlation coefficients change for one example species if these near-detection-limit values are excluded?

L515, Eq. 2: You need to define what the delta symbol means here, or above in Eq. 1

[Figure]

Table 1: Why is the GC-CIMS measurement "qualitative only"? And why is there no reference?

Figures S4 and S5 are not referenced in the main text

---

## Referee Comment (RC2) · Anonymous Referee #2 · 18 Dec 2017

Journal: ACP Title: Volatile organic compound emissions from biomass burning: identification, quantification, and emission factors from PTR-ToF during the FIREX 2016 laboratory experiment Author(s): Abigail R. Koss et al. MS No.: acp-2017-924 MS Type: Research article This work presents a comprehensive method to detect NMOGs released from laboratory simulated wildfires. The authors used different methods in parallel or in series with the PTR-ToF-MS to determine in confidence the emission ratios for different NMOGs from biomass burning. The paper is very well written and it shows and impressive amount of experimental work. Although the paper is more suitable to be published in Atmospheric Measurement Techniques however minor edits are required to be addressed by the authors before it is suitable to be published in ACP

1) Page 1, lines 29-31: the authors state that HCN is a compound that is not measure by PTR. A number of papers in the literature have already reported measurements using PTR-MS which the authors did not reference (Knighton et al 2009, Ambrose et al 2012 and Moussa et al 2016). In page 14, lines 428-430 with HCN proton affinity is very close to that of water did the authors use any RH correction when using FTIR to get sensitivities? How did the HCN sensitivity obtained from the FTIR method compares to the sensitivities of the other studies mentioned above? 2) Page 4, lines 115-116 what is the criterion that the authors used in order to divide stacks for different instruments? Is it based on fuel? Or did they select random stacks to test other techniques? 3) It was not clear throughout the paper if the authors did the corrections and calculations for emission factors for all the NMOGs from the different fuels using the different techniques 4) Page 8, line261-263 are the calculated and measured calibration factors in figure 1 obtained from this study or from literature it was not clear from the text 5) Page 11, lines 330 to 344 it is not clear why the authors used pyroles and butane nitrile as an example 6) Page 12, line 386-393 the authors started the paragraph with as shown in the Supplementary information, where exactly in SI? 7) Page 13, line 395 I think the authors meant monoterpene fragment and not isoprene fragment (C10H16OH+)

Overall a very good paper and I recommend its publication in ACP after the authors address the afore mentioned comments
* * *

---

## Author Comment (AC1) · 19 Jan 2018

Journal: ACP
Title: Volatile organic compound emissions from biomass burning: identification, quantification, and emission factors from PTR-ToF during the FIREX 2016 laboratory experiment
Author(s): Abigail R. Koss et al.
MS No.: acp-2017-924
MS Type: Research article

Response to reviewers
We thank both reviewers for their positive comments and helpful feedback. The manuscript has been revised accordingly. Our responses are written in blue text.

**Response to Reviewer 1**

This paper describes the identification of non-methane organic gases (NMOGs) measured by PTR-ToF mass spectrometry of biomass burning emissions during the 2016 FIREX experiment at the Missoula Fire lab. The emissions of the identified species are quantified and emission ratios and factors for the different species are derived. The measurements for individual organic gases are also compared with simultaneous measurements using other techniques, such as chemical ionization mass spectrometry (using I- and NO+) and infrared and UV spectroscopy. The authors find that while non-aromatic oxygenated compounds are the most abundant of the identified species, furans and aromatics comprise a large portion f the OH reactivity of the fresh emissions. This is a well-executed study on an important topic in atmospheric chemistry, as the amount of O3 and SOA attributable to fires is still highly uncertain, in part because of uncertainties in the emissions of NMOGs. In addition, the paper and the supplemental data go into a high level of detail on the measurements, calibrations, and subsequent calculations, making it very clear what was done, what assumptions were made, and the level of certainty for each of the provided values. The manuscript was also remarkably free of typos and other errors. I recommend publication after very minor revisions to address the few issues I noticed, listed below.

L183: What does the acronym "PLOT" mean here?

The acronym is now defined as "Porous Layer Open Tubular" at line 183.

L185: Who was the manufacturer of the "MXT-624" column? I'm assuming that's a catalogue number for some specific manufacturer?

The manufacturer (Restek) now indicated at line 185.

L327: How does the disagreement in identification discussed here potentially affect current review values of biomass burning emission factors?

This is a good question. We compared the newly identified compounds and compounds with revised identifications to the tables of EF provided in Akagi et al. (2011) and Yokelson et al. (2013).

To the text, at line 327, we have added:
        The compounds with new and revised identifications were compared to review values of emission factors in Akagi et al. (2011) and Yokelson et al. (2013). A limited number of species from PTR are included in these reviews, largely because of uncertainty in identification. PTR species that have

been detected but not included in review tables of EF include many more highly functionalized and larger molecules, and most of our updated identifications are these species. Yokelson et al. (2013) do include a number species from PTR (ion trap) that were not identified, and the identities of many of these have now been determined in this work.

Compounds that are included in review tables, and for which we have updated the assignment are mostly unsaturated hydrocarbons and heteroatom containing species, where the identifications have been updated to include other contributing VOCs. For such species, whose EF was determined solely from PTR, the actual emission factor should be lower than the reported value.

L327 and elsewhere: Please refer to the tab of Table S1 you are referring to, or make each tab of the Excel spreadsheet it's own table number (e.g., Table S1, S2, . . . or Table S1a, S1b, . . .)

We made each table its own table number (S1, S2, etc) and edited the text accordingly.

L420: This reference to Table 1 is probably redundant at this point.

Removed.

L445-447: My understanding is that the FTIR uncertainty in HONO is also very high, around 10 ppbv. Is this correct, and if so, does that affect the use of the FTIR to determine the PRT-ToF level of detection?

For the fire used to determine the PTR-ToF sensitivity (Fire 72), the standard deviation of the FTIR HONO measurement around zero was about 13 ppbv. The peak value of HONO in Fire 72 was around 800 ppbv. The PTR-ToF level of detection was determined from the slope between the PTR and FTIR measurements, which isn't affected significantly by the precision of the FTIR measurement at such high mixing ratios.

The slope (and PTR-ToF LoD) is dependent on the accuracy of the FTIR HONO measurement. The accuracy of the FTIR HONO is limited by the accuracy in the IR cross section. This uncertainty is <10%.

We have edited this section to read:

"HONO was sufficiently concentrated (800 ppbv max) in the fire, and the precision and accuracy of the FTIR HONO measurement were adequate to estimate a PTR-ToF 3σ LoD for HONO of about 9.5 ppbv. This is likely not sufficient to measure HONO in ambient air except in the most highly concentrated, fresh biomass burning plumes."

L457-459: Can you discuss how the slopes and correlation coefficients change for one example species if these near-detection-limit values are excluded?

According to the reviewer's suggestion, I redid the PTR-FTIR comparison for these five compounds, but removing FTIR data below a threshold of 1*LoD, and 3*LoD. This improves the slope and R2 for some compounds in some fires, but not consistently for any one compound, and overall R2 and slope are not improved. Therefore this statement is not supported as is. After further discussion it was determined that these species are typically harder to extract from FTIR spectra, and that some GC-CIMS measurements indicated furan could potentially have an interference on PTR. The text has been edited as follows:

"These species were often near the 0.73 Hz detection limit of the OP-FTIR and the discrepancy in slopes and low correlation coefficients are sometimes an effect of including this data in the intercomparison. Another reason for the disagreement may be that these species have either more interference or weaker spectral features than other compounds reported from FTIR. Furan may have an interference in PTR-ToF measurements of some fuels."

L515, Eq. 2: You need to define what the delta symbol means here, or above in Eq. 1

We added "Δ indicates the excess mixing ratio above background, as is explicitly written in Eq. 2."

Table 1: Why is the GC-CIMS measurement "qualitative only"? And why is there no reference?

Several calibrations of the GC-CIMS instrument were performed, but we ultimately decided that too few species were calibrated, the calibrations were not frequent enough, and the combined GC-CIMS instrument was not characterized well enough to report quantitative mixing ratios and detection limits. Instead, we compared the peak areas of each CIMS ion mass within individual chromatograms. There is no reference because this work is the first time this particular instrument has been described.

Figures S4 and S5 are not referenced in the main text.
Figures S3-S5 are referenced in the introductory paragraph of Section 3.3, and are now also referenced in the individual sections of section 3.3.

**Response to Reviewer 2**

This work presents a comprehensive method to detect NMOGs released from laboratory simulated wildfires. The authors used different methods in parallel or in series with the PTR-ToF-MS to determine in confidence the emission ratios for different NMOGs from biomass burning. The paper is very well written and it shows and impressive amount of experimental work. Although the paper is more suitable to be published in Atmospheric Measurement Techniques however minor edits are required to be addressed by the authors before it is suitable to be published in ACP

1) Page 1, lines 29-31: the authors state that HCN is a compound that is not measure by PTR. A number of papers in the literature have already reported measurements using PTR-MS which the authors did not reference (Knighton et al 2009, Ambrose et al 2012 and Moussa et al 2016). In page 14, lines 428-430 with HCN proton affinity is very close to that of water did the authors use any RH correction when using FTIR to get sensitivities? How did the HCN sensitivity obtained from the FTIR method compares to the sensitivities of the other studies mentioned above?

Line 31 states that HCN is not frequently reported from PTR. We did correct for humidity-dependent sensitivity of HCN. We generated a steady concentration of HCN and sampled it with our PTR at a wide range of humidity. The reported data are corrected by this curve. Ambrose et al. 2012 report measurements of acetonitrile from PTR, but HCN from PTR is not reported. Moussa et al. 2016 do not report the sensitivity to HCN. Knighton et al. 2009 report a sensitivity of about 3 cps/ppbv HCN. The instrument used in this study has a sensitivity of about 106 cps/ppbv HCN.

In section 3.3.1, which discusses comparison of HCN between FTIR and PTR, we have added:
"Sensitivity to HCN has strong humidity dependence (Knighton et al., 2009; Moussa et al., 2016), and this was experimentally determined and corrected."

2) Page 4, lines 115-116 what is the criterion that the authors used in order to divide stacks for different instruments? Is it based on fuel? Or did they select random stacks to test other techniques?

We have inserted an explanation at this location:
"The particular fires measured with each technique were selected as follows. At least one fire of each fuel type was measured directly with PTR-ToF, and coniferous fuels were measured at least twice. Given these restrictions with the PTR-ToF measurement, the widest possible range of fuel types was measured with NO+-CIMS. GC-CIMS stack burns were selected for longer-burning fuels that allowed collection of more than one sample. GC-CIMS room burns were selected to explore a range of fuel types."

3) It was not clear throughout the paper if the authors did the corrections and calculations for emission factors for all the NMOGs from the different fuels using the different techniques

At line 116 we have added,
"We were not able to measure every fuel type with every instrumental technique."

4) Page 8, line261-263 are the calculated and measured calibration factors in figure 1 obtained from this study or from literature it was not clear from the text

We have clarified this as,

"Figure 1 compares the measured (from this work) and calculated (using the method from Sekimoto et al., 2017a) calibration factors for several compounds."

5) Page 11, lines 330 to 344 it is not clear why the authors used pyroles and butane nitrile as an example

Pyrroles and butane nitriles are isomers and cannot be distinguished with on-line PTR-MS. We have clarified this section to read,
"These isomers cannot be distinguished by online PTR-ToF, and each contributes a different amount to the total signal of m/z 68.050 $C_4H_5NH^+$. Panel C shows the same 280-second retention-time period, from a sample taken immediately after the one shown in Panel B, but measured with $NO^+$-CIMS. These isomers can be identified by comparing GC-PTR-ToF and GC-$NO^+$-CIMS chromatography, as $NO^+$ reacts with pyrrole but not nitriles. The GC retention time, when it is known for a particular compound, provides additional support for the identification."

6) Page 12, line 386-393 the authors started the paragraph with as shown in the Supplementary information, where exactly in SI?

Table S7. This is now added to the text.

7) Page 13, line 395 I think the authors meant monoterpene fragment and not isoprene fragment (C10H16OH+)

Isoprene fragment is correct. The ion in question is m/z 67 $C_5H_7^+$. This ion is detected when both cyclopentadiene and isoprene elute from the GC.

Overall a very good paper and I recommend its publication in ACP after the authors address the afore mentioned comments

Thank you!

**Additional corrections**

During the Discussions phase of this manuscript, several minor errors were indicated to us through private communication. The corrections are described below:

1) The citation of Selimovic et al. (submitted) has been updated to:
   Selimovic, V., Yokelson, R. J., Warneke, C., Roberts, J. M., de Gouw, J., Reardon, J., and Griffith, D. W. T.: Aerosol optical properties and trace gas emissions by PAX and OP-FTIR for laboratory-simulated western US wildfires during FIREX, Atmos. Chem. Phys. Discuss., acp-2017-859, in review, 2017.

2) In Table 1, a citation of Selimovic et al. (2017) is added to the Reference for OP-FTIR.

3) The reference to Sekimoto et al. 2017b has been updated to:
   Sekimoto, K., Koss, A., Yuan, B., Coggon, M. M., Selimovic, V., Warneke, C., Yokelson, R., and De Gouw, J.: High- and low-temperature pyrolysis profiles describe volatile organic compound emissions from western US wildfire fuels, in preparation, 2018.

4) The compound at m/z 117 $C_5H_8O_3H^+$ was labeled "2,5-dihydroxymethyl dihydrofurfural" and should have been labeled "5-hydroxymethyl tetrahydro 2-furanone/5-hydroxy tetrahydro 2-furfural". This is corrected in Table 2 and in the supplemental tables. No calculations of OH reactivity, volatility, or figures are affected.

5) In Table S2 the labels for unknown compounds were disordered. This has been corrected and the format of Tables S2 and S3 has been adjusted to be the same (identified compounds first, followed by unidentified compounds). The "NMOG identity" label for species in Tables S2 and S3 has been adjusted so that the text matches exactly Table S1 (e.g. "C8 nitrile" –> "C8 nitriles"; "H2S" -> "hydrogen sulfide")

6) In Table S3 the columns were labeled "Average ER" and this has been corrected to "Average EF".

References

Akagi, S. K., Yokelson, R. J., Wiedinmyer, C., Alvarado, M. J., Reid, J. S., Karl, T., Crounse, J. D., and Wennberg, P. O.: Emission factors for open and domestic biomass burning for use in atmospheric models, Atmos. Chem. Phys., 11, 4039-4072, https://doi.org/10.5194/acp-11-4039-2011, 2011.

Yokelson, R. J., Burling, I. R., Gilman, J. B., Warneke, C., Stockwell, C. E., de Gouw, J., Akagi, S. K., Urbanski, S. P., Veres, P., Roberts, J. M., Kuster, W. C., Reardon, J., Griffith, D. W. T., Johnson, T. J., Hosseini, S., Miller, J. W., Cocker III, D. R., Jung, H., and Weise, D. R.: Coupling field and laboratory measurements to estimate the emission factors of identified and unidentified trace gases for prescribed fires, Atmos. Chem. Phys., 13, 89-116, https://doi.org/10.5194/acp-13-89-2013, 2013.

---

## Author Response (AR1)

**For the best experience, open this PDF portfolio in Acrobat X or Adobe Reader X, or later.**